



# Estimating global land system impacts of timber plantations using MAgPIE 4.3.2

Abhijeet Mishra[1,2], Florian Humpenöder[1], Jan Philipp Dietrich[1], Benjamin Leon Bodirsky[1], Brent Sohngen[3], Christopher P. O. Reyer[1], Hermann Lotze-Campen[1,2], and Alexander Popp[1]

[1]Potsdam Institute for Climate Impact Research (PIK), Member of the Leibniz Association, P.O. Box 60 12 03, 14412 Potsdam, Germany
[2]Humboldt University of Berlin, Department of Agricultural Economics, Unter den Linden 6, 10099 Berlin, Germany
[3]The Ohio State University Department of Agricultural, Environmental and Development Economics, Columbus, Ohio, United States of America.

**Correspondence:** Abhijeet Mishra (mishra@pik-potsdam.de)

**Abstract.**

Out of 1150 Mha of forests designated primarily for production purposes in 2020, plantations account for 11% (131 Mha) of area and fulfilled more than 33% of the global industrial roundwood demand. Yet, adding additional timber plantations to meet increasing timber demand increases competition for scarce land resources between different land-uses for food, feed, livestock and timber production. Despite their significance in roundwood production, the importance of timber plantations in meeting the long-term timber demand and the implications of plantation expansion for overall land-use dynamics have not been studied in detail so far, in particular not the competition for land between agriculture and forestry in existing land-use models.

This paper describes the extension of the modular, open-source land-system Model of Agricultural Production and its Impact on the Environment (MAgPIE) by a detailed representation of forest land, timber production and timber demand dynamics. These extensions allow for understanding the land-use dynamics (including competition for land) and associated land-use change emissions of timber production.

We show that the spatial cropland patterns differ when timber production is accounted for, indicating that timber plantations compete with cropland for the same scarce land resources. When plantations are established on cropland, it causes cropland expansion and deforestation elsewhere. As a result of increasing timber demand, we show an increase in plantations area by 140% until the end of the century (+132 Mha in 1995-2100). We also observe in our model results that the increasing demand for timber increases scarcity of land, and causes intensification through yield increasing technological change by 117% in croplands by 2100 relative to 1995. Through the inclusion of new forest plantation and natural forest dynamics, our estimates of land-related $CO_2$ emissions match better with observed data in particular the gross land-use change emissions and carbon uptake (via regrowth), reflecting higher deforestation for expansion of managed land and timber production, and higher regrowth in natural forests as well as plantations.



## 1 Introduction

Forests cover 4060 million hectares (Mha) of the global land (31%) in 2020. Out of this 4060 Mha, 1110 Mha are primary, 2657 Mha are secondary and 293 Mha are planted forests of which plantations cover 131 Mha and other planted forests cover 162 Mha, based on FAO (2020a) definitions. According to FAO (2020a), 1150Mha of forest are designated as production forests.

Plantations, as a very special forest land-use type according to FAO definitions, account for 11% of that area (and only 3% of global forest area) but likely supply more than 33% (560 $Mm^3$) of global roundwood demand (1683 $Mm^3$). This relatively large contribution compared to the area covered underlines plantations' special role in global land use dynamics. Roundwood consists of two sub-categories, industrial roundwood and wood fuel.

Historical trends show a continuous increase in the share of roundwood production coming from plantations (Jürgensen
et al., 2014). This trend indicates the efficacy and importance of timber plantations in meeting roundwood demand and the role of renewable forest management in natural forests especially in North America and Europe (Siry et al., 2018; Biber et al., 2020). The remaining share comes from other sources including harvesting of natural forests or managed secondary or planted forests. Deforestation continues to occur at a large scale with wood harvesting being an important driving factor after cropland expansion (Curtis et al., 2018).

Deforestation contributes to about a third (3.8 Gt $CO_2$ $yr^{-1}$) of Agriculture, Forestry and Land-Use (AFOLU) change emissions (10-12 Gt $CO_2$ $yr^{-1}$) (Jia et al., 2019; Smith et al., 2014), and as it is an important driver of biodiversity loss, a better understanding of how we can produce timber using land resources efficiently is imperative. Because of their higher productivity as compared to natural forests (FAO, 2013), timber plantations have the potential to fulfill a major portion of global roundwood demand while using a relatively small amount of land. Yet, assuming land distribution among different land-uses
to be a zero-sum game, higher demand for timber plantation areas has to come from reducing other land uses (e.g. agriculture or natural vegetation). This creates additional pressures on the land-system.

Land being a limited resource drives competition between land-uses due to increasing food, feed and timber demand. Demand for roundwood and the way this roundwood is produced drives competition for land via more forest area which competes for demand for land with agriculture. Land-use models can help in analyzing these land competition dynamics based on ob-
served data by optimizing a set of objective(s) and minimizing negative trade-offs between land uses (Verhagen et al., 2018). Understanding such competition helps to reveal how changes in the land system affect the functioning of the land system as a whole and the trade-offs this competition may entail (Crate et al., 2017).

As part of land systems, forest resource use has been included in many modeling activities including Integrated Assessment Models (IAMs) like the Global Change Analysis Model (GCAM) (Calvin et al., 2019; Wise et al., 2014) and the Integrated
Model to Assess the Global Environment (IMAGE) (Stehfest et al., 2014). Forests are also included in varying degrees of representation in recursive dynamic optimization models like the Global Forest Sector Model (EFI-GTM) (Kallio et al., 2004) and the Global Biosphere Management Model (GLOBIOM) (Havlík et al., 2011) coupled with the Global Forest Model (G4M) (Kindermann et al., 2006). Timber supply and demand are also represented in the Global Timber Model (GTM) (Sohngen et al.,





1999) which is an inter-temporal optimization model. A detailed review of recent developments and applications of partial
equilibrium models in the forest sector is provided by Latta et al. (2013). Yet, existing land-use models or forest economics
models at higher spatial resolution either simulate detailed forest types and neglect competition for land or vice-versa. No
existing land-use model to our knowledge combines both of these features at the global scale.

To correctly represent the competition for land and the role of different forest types in meeting growing roundwood demand,
ideally, a land-use model should a) represent land resource competition while accounting for food, feed and timber demand,
and, b) represent different growth rates between natural and planted forests (with accounting of optimal rotations in timber
plantations).

Yet, out of the recursive dynamic models mentioned above, partial equilibrium models like EFI-GTM and GTM do not use
spatially explicit differences in forest growth rates but use aggregated forest inventory data as model inputs. Both of these
models rather focus on detailed representation of the forest and timber industry with great detail but do not model competition
for land between forests and agriculture at a fine spatial scale. IMAGE and GLOBIOM, both use spatially explicit differences in
forest growth rates and tree species while representing competition for land between forests and agriculture but do not explicitly
differentiate between natural forests and timber plantations. In IMAGE, land-use evolution for timber plantations is a model
parameter and is not endogenously determined. GLOBIOM when coupled with G4M also circumvents the myopic nature of
recursive dynamic models as G4M results are linked to GLOBIOM for making appropriate land-use change decisions regarding
wood production and forest land-use. GCAM models competition between land-uses via land competition nests (Snyder et al.,
2020) where land-use categories belonging to the same category in the nest (e.g. crops) are assumed to compete more directly
with each other than with land-uses in other category (e.g. forest) (van de Ven et al., 2021).

In light of these limitations of representing timber plantations in the land-use modeling frameworks described above, tools
that quantify and analyze land competition while explicitly accounting for the specifics of forest plantations within a uniform
modeling framework are required. The Model of Agricultural Production and its Impact on the Environment (MAgPIE) uses
both biophysical and economic drivers to simulate land-use change and its impact on the environment while accounting for
feed, food and livestock demand (Popp et al., 2010; Lotze-Campen et al., 2008; Dietrich et al., 2019; Bodirsky et al., 2020).
Driven by the motivation to represent coherent forest land-use dynamics within a single modeling framework, we present here
an extension of the MAgPIE 4 modeling framework by timber production and associated land-use dynamics. The extension
addresses the forestry sector modeling gaps outlined above via new MAgPIE modules that differentiate timber plantations and
natural vegetation land-use.





## 2 Methods

### 2.1 Model description

#### 2.1.1 MAgPIE framework

The MAgPIE modeling framework (Dietrich et al., 2019; Lotze-Campen et al., 2008) is a global multi-regional land system model. The objective function of MAgPIE is to minimize the global costs to produce food, feed, bioenergy and timber throughout the 21st century in recursive dynamic mode with limited foresight. In real-world, when we usually do not have absolute certainty in what the future holds, and provided the long time horizons in the establishment of new trees today followed by harvesting such trees sometime in the future calls for using a recursive-dynamic model for understanding how today's decisions

impact tomorrow's behaviour. MAgPIE is driven by demand for agricultural commodities and roundwood, which is calculated based on population and income projections for the 21st century from the Shared Socioeconomic Pathways (SSPs).

MAgPIE derives specific land-use patterns, yields and total costs of agricultural and roundwood production for each simulation cluster as described in Dietrich et al. (2019). MAgPIE's optimization is bound by spatially explicit biophysical constraints derived from the global gridded crop and hydrology model LPJmL (Bondeau et al., 2007). For this assessment, the spatially

explicit (0.5° resolution) LPJmL outputs are aggregated for MAgPIE into 200 simulation units/clusters using a clustering algorithm (Dietrich et al., 2019, 2013) as shown in fig. 1. MAgPIE is a non-linear mathematical programming model written in GAMS and solved with CONOPT4 solver.

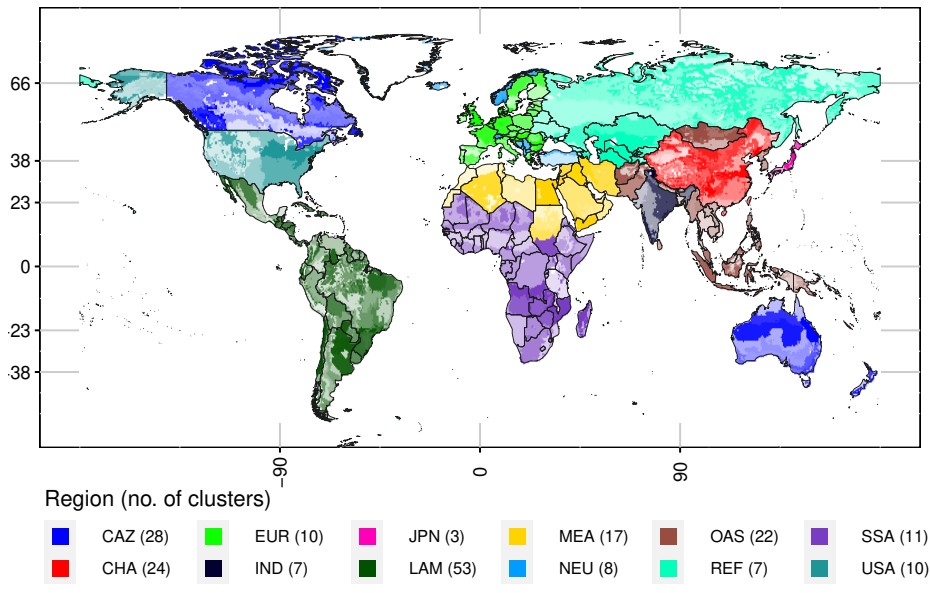

**Figure 1.** 200 Simulation clusters in MAgPIE based on Dietrich et al. (2020a) on a 0.5° resolution grid. Clusters in each region are plotted on a gradient from darkest to lightest shade of color representing a region.





### 2.1.2 MAgPIE 4.3.2

The existing MAgPIE 4 framework (Dietrich et al., 2019) has been extended by the inclusion of timber production via forest

land and timber demand, which we refer to as MAgPIE 4.3.2 in the text. Growth function for forests (Humpenöder et al., 2014) are parameterized by using plantation and natural vegetation specific parameters from Braakhekke et al. (2019). Finally, the trade representation was also extended to include industrial roundwood and wood fuel trade. The extension of the MAgPIE framework from version 4 to version 4.3.2 is shown in fig. 2.

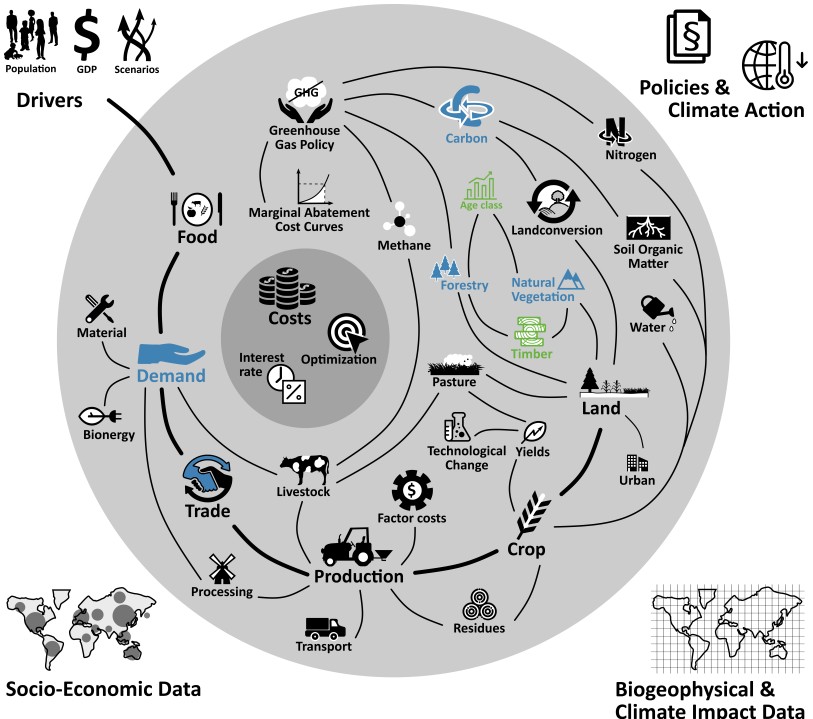

**Figure 2.** Extended MAgPIE 4.3.2 framework. Blue color represents update to existing modules, green color represents new inclusions to Dietrich et al. (2019). See the model documentation (Dietrich et al., 2020b) for a more detailed presentation of module interactions and their implementations.

### 2.2 Scenarios

We analyse two scenarios here namely *default* and *forestry* (Table 1). Both, default and forestry scenarios take assumptions from the SSP2 storyline also known as *business as usual* or *middle of the road* scenario (Riahi et al., 2017). In the default case, we replicate assumptions from a standard MAgPIE configuration based on Dietrich et al. (2020b), where a) Timber demand is not modeled, b) No forest is harvested for timber production, c) No competition for land between agriculture and forestry, and d) Secondary forests and plantations are assumed to belong to the highest age-class during model initialization. The setup of



the default scenario without wood demand, no harvest from plantations (and other forests) and no new plantation establishment
implies that the plantation area remains constant at 1995 levels.

The forestry scenario on the other hand accounts for a) GDP and population-driven industrial roundwood and wood fuel demand, b) Plantations and natural forests as source of timber production, c) Endogenous competition for scarce land resources between agriculture and forestry, and d) Heterogeneous age-class structure of secondary forests and plantations during initial-

ization. Plantation forests are initialized such that there is a higher weight provided to younger age-classes reflecting the notion that replanting has continued to exceed harvests in plantations in the last decades. Secondary forests are initialized based on the land distribution among age-classes described in Poulter et al. (2019).

**Table 1.** Summary of main differences between scenario setups.

|  | Food demand | Feed demand | Timber demand | Source of timber production | Competition (agriculture vs. forestry) | Initial state of forests | Plantation area |
|---|---|---|---|---|---|---|---|
| Default | Yes | Yes | No | No | No | Homogeneous | Static |
| Forestry | Yes | Yes | Yes | Yes | Yes | Heterogeneous | Dynamic |

## 2.3 Forestry rotation lengths

According to the von Thünen-Jevons single rotation-period model described in Amacher et al. (2009), the economically optmial

time to harvest trees occurs when the Instantaneous Growth Rates (IGR) is equivalent to the interest rate in the economy (equation 1). For our implementation, we use region-specific interest rates (Table A2) and assume that each cluster belonging to a region has the same prevailing interest rate to which the IGR is compared to. Relationship between MAgPIE regions and clusters is described in Dietrich et al. (2019) and country to MAgPIE region mapping is provided in Table A1.

$$\frac{f_{j,ac}}{f'_{j,ac}} = r_j \tag{1}$$

In equation 1, $f_{j,ac}$ is the cluster level ($j$) age-class ($ac$) specific carbon density and $f'_{j,ac}$ is the first derivative of the same with respect to age-classes. $r_j$ is the cluster level interest rate in the economy (assuming every cluster in a given region has the same prevailing interest rate). Instead of using forest volume described in Amacher et al. (2009), we use carbon density as a proxy for the same. Long term average potential carbon density information for each MAgPIE cluster is obtained from LPJmL (Bondeau et al., 2007). This carbon density information is fed into a Chapman-Richard's growth function to derive age-class

specific carbon densities i.e. $f(ac)$ based on Humpenöder et al. (2014) (fig. 3a). The first derivative of these carbon densities provides the marginal values with respect to age classes (fig. 3b). The ratio between the original and marginal carbon densities provides the IGR i.e., $\frac{f_{j,ac}}{f'_{j,ac}}$ (fig. 3c). Equating IGR to interest rates ($r_j$) provides the cluster specific optimal rotation lengths (fig. 3d) i.e., the optimal age-class at which harvest of timber plantation is allowed in each cluster. Rotation length decisions

once made cannot be altered later during optimization in MAgPIE. Spatially explicit rotation lengths calculated in MAgPIE

are shown in fig. 4 based on the assumed interest rates.

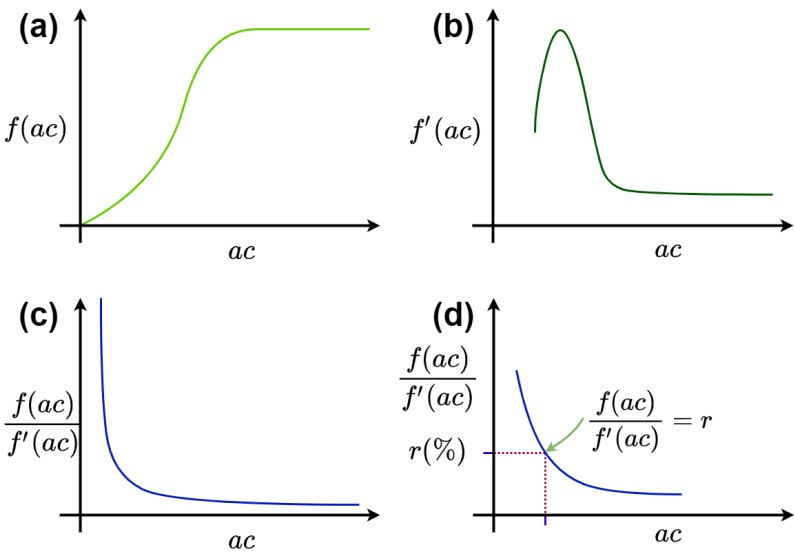

**Figure 3.** Qualitative representation of rotation length calculation using single rotation model in MAgPIE based on Amacher et al. (2009). The x-axis represents the age-class equivalent of rotation lengths. a) S-shaped growth curve calculation for every MAgPIE cluster, b) First derivative of these cluster-specific carbon densities, c) Ration of original and marginal carbon densities, d) Equating IGR with interest rates.

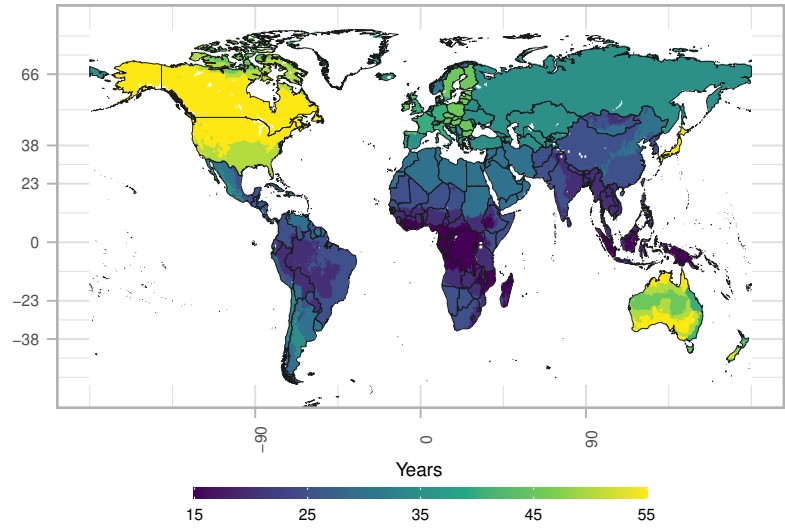

**Figure 4.** Spatially explicit regional rotation lengths in MAgPIE (years). Plantations belonging to cells with the same color have the same rotation length.





## 2.4 Forest initialization

In MAgPIE, forestry rotation lengths dictate what the initial distribution of planted forest area should look like in 1995. The country-level planted forest area from FAO (2015) is downscaled to a 0.5° grid using area-weighted mean of wood removals (Hurtt et al., 2018) and then upscaled to MAgPIE cluster level (Dietrich et al., 2019) for initialization of 1995 values. Distribu-
tion of this area among different age classes i.e., the age-class structure in plantations during initialization is driven by rotation lengths. Aggregated cluster level planted forest area is distributed first between plantations and other plantation areas based on historical share of such distinction based on FAO (2020b). Cluster level plantation area is then divided among age-classes such that there is a higher weight provided to younger age-classes reflecting the notion that plantation area establishment has increased in the last decades. Figure 5 shows the initialization of the MAgPIE plantation area in each cell in 1995.

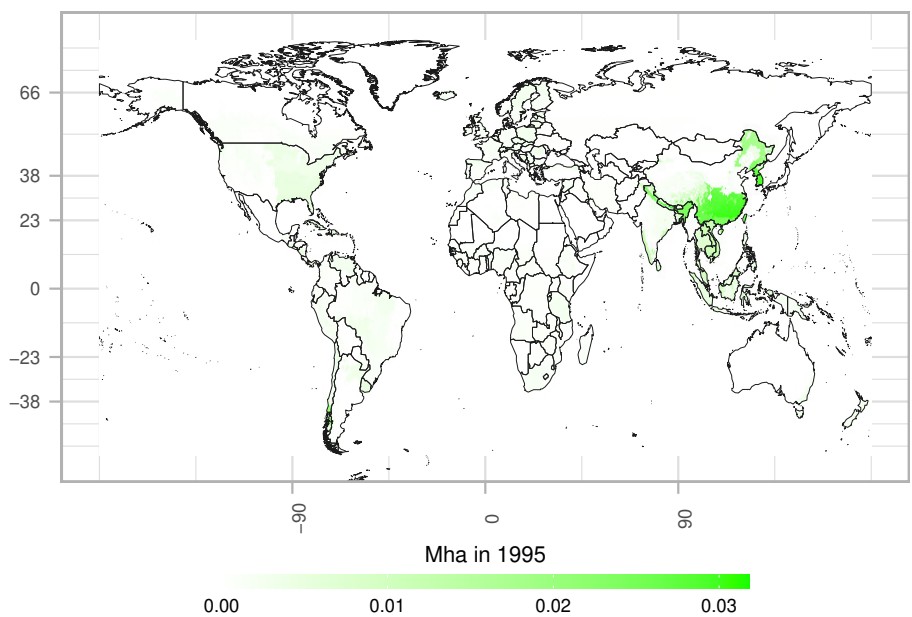

**Figure 5.** Initialization of plantation area in 1995 in forestry scenario using rotation length for age-class distribution (Mha)

Natural vegetation in MAgPIE consists of primary forest (untouched pristine forest without signs of human intervention), secondary forests (forests with some indication of human intervention and management) and other land (degraded forests or uncultivated land with lower vegetation carbon density than normal forests). The initial spatial distribution of the natural vege­tation in MAgPIE is based on the Land-Use Harmonization (LUH) data set (Hurtt et al., 2018) and adjusted for harmonization with FRA reported data (MacDicken, 2015) with re-allocation of natural vegetation area. The area allocated to primary forests
is assumed to exist in the highest age-class in 1995. Area allocated to secondary forests is assumed to follow the distribution of forests in different age-classes based on Poulter et al. (2019). After the initialization of forest areas, the development of





forest-cover is modeled endogenously in the model and driven by roundwood demand, timber harvest costs, expected yields, carbon prices, demand for agricultural land, land-use change costs and land-use change constraints.

## 2.5  Timber demand

Demand for end-use wood products in MAgPIE is driven by changes in per capita income and population for the Shared Socioeconomic Pathway 2 (SSP2) storyline. Here we take assumptions from the SSP2 storyline to derive the timber demand. We use a simple demand function specification from Lauri et al. (2019), initialized with historical demand volumes from FAOSTAT (FAO, 2017) and shifted over time using changes in GDP and population as shown in equation 2. The demand estimates for roundwood, Industrial roundwood, Wood fuel, Other Industrial roundwood, Pulpwood, Sawlogs and Veneer logs,

Fibreboard, Particleboard and OSB, Wood pulp, Sawnwood, Plywood, Veneer sheets, Wood-based panels and Other sawn wood are made independently in the model.

$$Q_{t+1,wp} = Q_{t,wp} * \frac{N_{t+1}}{N_t} * \left( \frac{I_{t+1}}{I_t} \right)^{E_{wp}} \tag{2}$$

Here, $t$ is the simulation time step i.e. time and $wp$ are different demand categories for wood products. $Q$ is the annual timber demand in $Mm^3$. $N$ is population and $I$ is income in USD per capita per year (in Purchase Power Parity (PPP), base

2005). $E$ is the income elasticity of wood products based on Morland et al. (2018). End-use wood product demand calculated from equation 2 is aggregated and used as a demand for two wood products - industrial roundwood and wood fuel. Industrial roundwood demand is calculated as the sum of Fibreboard, Particleboard and OSB, Plywood, Veneer sheets, Wood pulp, Sawnwood, Other sawn wood and Other Industrial roundwood. The processing of wood products is not explicitly modeled in MAgPIE. By-products of end-use production activities and recycling of roundwood is also not accounted for in MAgPIE.

Wood fuel is assumed to come from two different sources: direct harvest and logging residues from harvesting for industrial roundwood.

Global industrial roundwood and wood fuel demand modeled in MAgPIE is shown in fig. 6 along with validation from historical data reported by FAO (regional numbers in fig. A4). Wood fuel enters demand calculations with a negative income elasticity based on Morland et al. (2018) to be consistent with the decreasing residential sector biomass use for energy in an

SSP2 world (Lauri et al., 2019; IIASA, 2018). We use the logging residue data from Oswalt et al. (2019) indicating that 30% of industrial roundwood harvest is residue. Assuming 50% of this is recovered from forests (Pokharel et al. (2017) report a range of 30-70% from available literature), we use a maximum of 15% of biomass removed during industrial roundwood production as wood residues which can contribute towards fulfilling wood fuel demand.



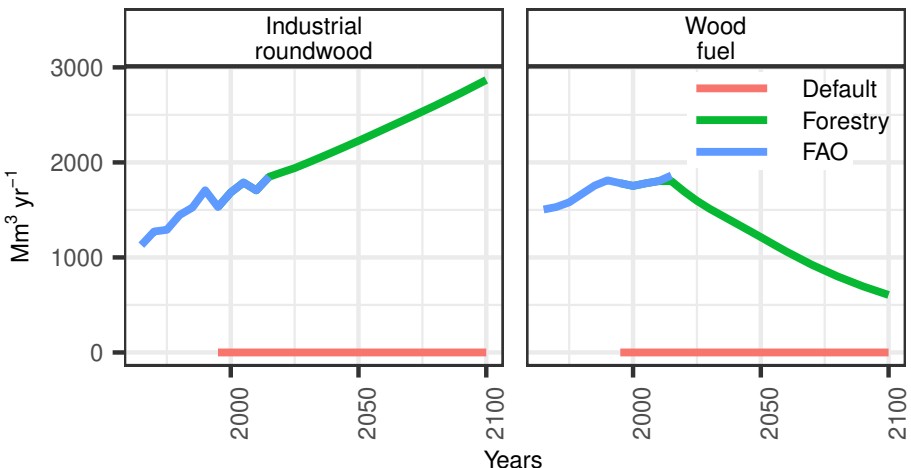

**Figure 6.** Global industrial roundwood and wood fuel demand for 1995-2100 (Mm³ yr⁻¹). Historical data from FAO is based on FAO (2017). MAgPIE output is for model run using the forestry scenario.

## 2.6 Forest biomass

Biomass which can be potentially removed from natural forests is calculated based on the average long-term vegetation carbon densities in natural vegetation from LPJmL. Growth of natural vegetation in MAgPIE follows an s-shaped growth curve as described in Humpenöder et al. (2014), but with updated growth curve parameters based on Braakhekke et al. (2019). Timber plantations on the other hand are considered more productive (for a younger stand age per unit area) compared to primary forests and secondary forests (Del Lungo et al., 2006). To reflect this, we use a different parametrization of the timber plantation growth

function as compared to natural forests based on Braakhekke et al. (2019). Harvestable biomass from forests are calculated as shown in equation 3 based on Ravindranath and Ostwald (2007) and Standard (2013).

$$yr_{\text{t,j,ac,ft}} = \frac{C_{\text{t,j,ac,ft}} * r_{\text{ft}}}{cf * \sum_{clcl} (kg_{\text{j,clcl}} * b_{\text{j,ac,clcl}})} \tag{3}$$

Here, $t$ is the simulation step i.e. time, $j$ is the MAgPIE simulation cluster, $ft$ is the forest type i.e., plantations or natural vegetation. $ac$ is the forest age-class, $clcl$ is the Köppen-Geiger climate class. $y$ is the age-class ($ac$) specific biomass yield in

tDM/ha, $C$ is the carbon density in tC/ha, $r$ is shoot-to-root ratio, $cf$ is the carbon fraction in dry matter (IPCC, 2019), $kg$ is the Köppen-Geiger climate classification (Rubel and Kottek, 2010) and $b$ is the biomass expansion factor (FAO, 2013). Forest classification in MAgPIE is represented in fig. 7 and the detailed description of forest land dynamics are described in Dietrich et al. (2020a).





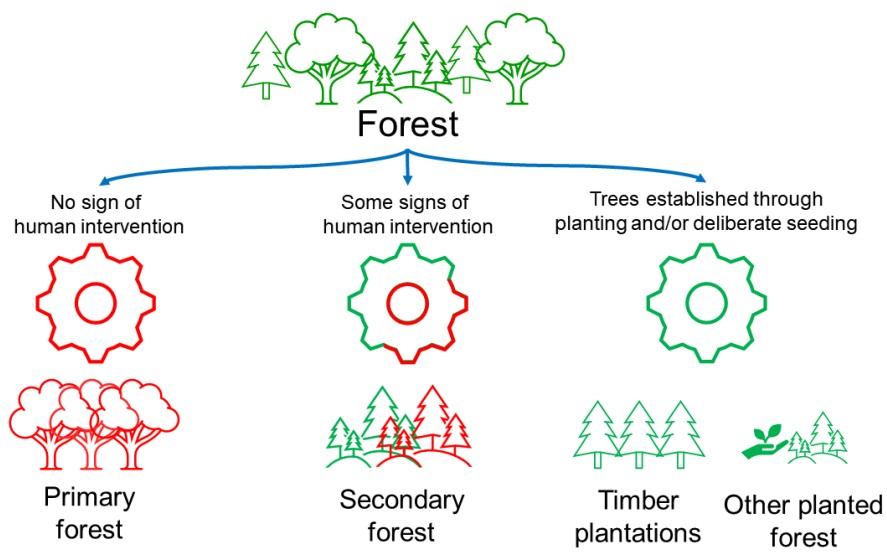

**Figure 7.** Forest classification in MAgPIE built on FAO (2015) definitions and classification

The carbon density in plantations and natural forests is calibrated using a scaling factor to match historically reported forest
area at regional level (FAO, 2020a). This scaling factor is calculated as the ratio between observed growing stocks (both,
in plantations and natural forests) reported by FAO (2020a) and initialized growing stocks in MAgPIE before optimization.
Calibrated growing stock in natural forests and plantations at global level is shown in fig. 8 (regional numbers shown in fig.
A8).

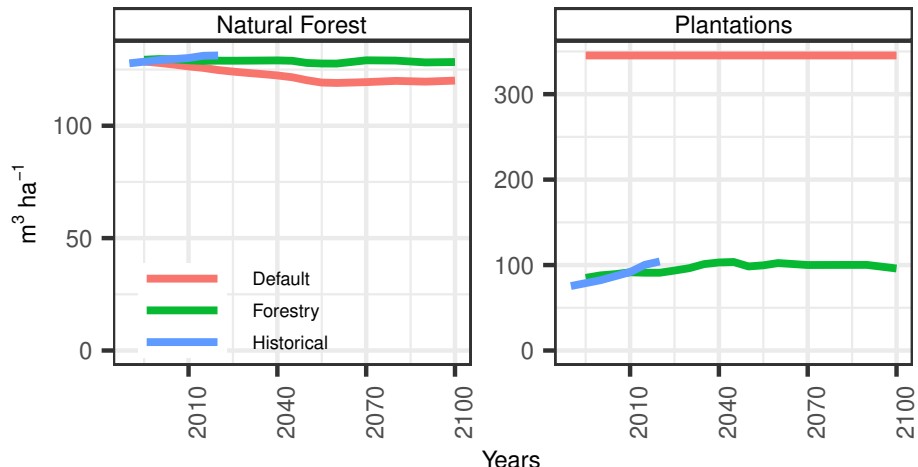

**Figure 8.** Global growing stock in natural forests and plantations between 1995-2100 (m$^3$ ha$^{-1}$). Historical values are taken from FAO
(2020b).





### 2.7 Timber production

#### 2.7.1 Plantation establishment

Amount of newly established timber plantations depends on current roundwood demand and expected future yields. Expected future yields in plantations are calculated based on the rotation lengths. As shown in equation 4, we define a regional constraint while establishing new timber plantations.

$$\sum_{j,ac}^{i} plant_{j,ac'} * y_j \geq \sum_{rw} Q_{i,rw} * \sigma_{i,rw} * \eta_i * ES_i \tag{4}$$

Here, *plant* is the plantation land, *j* is the MAgPIE simulation cluster, *ac'* is the age classes to be established (usually the youngest age-class that is *ac0*), $Q_{i,rw}$ is the regional annual demand for roundwood (*rw*) i.e., industrial roundwood and wood fuel in region *i* as shown in fig. 6. $\sigma_{i,rw}$ is the regional self-sufficiency ratio of roundwood (industrial roundwood and wood fuel) production (Table A3), $\eta_i$ is the share of production which can come from plantations based on extrapolations from Pöyry (1999). For the extrapolation of these shares, we assume (starting from last historically available data in 2000), 1% increase per annum till 2020, 0.4% increase per annum between 2020-2050 and 0.2% increase from 2050-2100 (Table A4). $ES_i$ is a calibration factor to nudge the model towards historical plantation area patterns (Table A5) via establishment of new plantations.

For example, Assuming industrial roundwood demand of 100 Mm3 in 2020 in region *i* with self-sufficiency ratio of 0.8 and $\eta_i$ of 0.5, the model will need to establish plantations such that 100 * 0.8 * 0.5 = 40 Mm3 of timber can be produced from this region in the future. The model then tries to establish new plantations in the simulation step depending on expected yields. Assuming this region has 2 clusters, both with an expected yield of 5 m$^3$ ha$-1$, there will be 4 Mha ((1/2)*40/5) of plantations established in each cluster i.e, 8 Mha of total new plantations in this region.

#### 2.7.2 Timber harvesting

Timber plantations are harvested once they reach maturity at the specified optimal rotation lengths. After every time step, forest age classes are shifted forward. Plantations are protected from harvest during the whole duration of time below their specified rotation length. There is no such restriction on the harvest of natural vegetation based on age and maturity. Roundwood (for industrial roundwood and wood fuel) can be produced from both natural forests (primary and secondary forests) and from managed plantations (forestry), which we distinguish according to figure 7. Additionally, wood fuel can also be harvested from *other* land, which is defined as non-managed land that has an insufficient carbon stock (<20 tC ha$^{-1}$) to be classified as forest. Timber production from forests is calculated based on the area harvested and the harvestable yields (3).

### 2.8 Land-use change emissions

Net CO$_2$ flux from land-use, land-use Change and Forestry (LULUCF) includes CO$_2$ fluxes from forest harvest (for roundwood production), deforestation (clearing forest for alternative land-use), afforestation, shifting cultivation (deforestation followed





by abandoning) and regrowth of forests following wood harvest or abandonment. Some of these activities lead to emissions of

$CO_2$ to the atmosphere (burning wood fuel after harvest, conversion of forests to agricultural land), while others lead to $CO_2$ sinks (afforestation, regrowth, long term carbon stored in harvested wood products).

Land, in particular biomass production from vegetation, affects both the source and sinks of $CO_2$. While reporting on LULUCF emissions, usually the long term carbon stored in wood products is either not reported or not accounted for in models which simulate forest land-use (Stehfest et al., 2019; Havlík et al., 2011; Braakhekke et al., 2019; Doelman et al., 2018, 2020;

Humpenöder et al., 2018). As management of forests and different uses of harvested wood play a crucial role in the regulation of the concentration of atmospheric $CO_2$, it is important to account for this pool while reporting LULUCF emissions (IPCC, 2019; Johnston and Radeloff, 2019; Böttcher and Reise, 2020; Zhang et al., 2020).

In MAgPIE we account for gross land-use change emission (i.e. land-use change emissions not including regrowth), emissions due to shifting agriculture (as part of gross land-use change emissions) based on historically observed deforestation driver

rates from (Curtis et al., 2018), regrowth in forests and other land as well as long term carbon storage in wood products while also calculating the slow release of $CO_2$ back into the atmosphere from these wood products due to decay (fig. 9). Carbon stored in harvested wood products (HWPs) can affect national greenhouse gas (GHG) inventories, in which the production and end-use of HWPs play a key role (Johnston and Radeloff, 2019). We account for this long term carbon storage in wood according to the guidance provided by The Intergovernmental Panel on Climate Change (IPCC) as defined in equation 5 (IPCC,

245   2019).

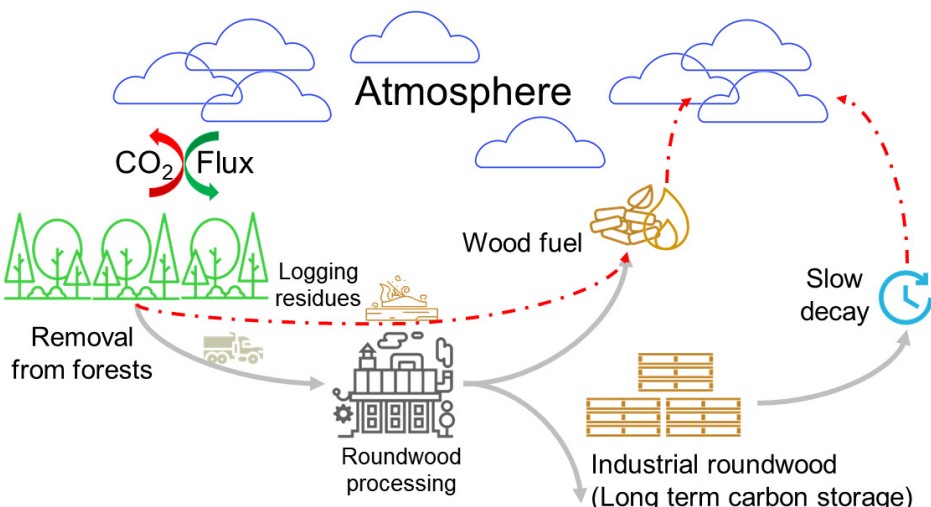

**Figure 9.** Concept for accounting for carbon emission and storage dynamics from forests and harvested roundwood. Wood fuel is assumed to be emitted within the optimization step in which it is harvested. Industrial roundwood enters a long term storage pool, from which slow turnover happens and is tracked via IPCC (2019) methodology described in equation 5.





$$C_{t+1} = e^{-k} * C_t + \left[ \frac{(1 - e^{-k})}{k} \right] * inflow_t \tag{5a}$$

$$\Delta C_t = C_{t+1} - C_t \tag{5b}$$

$inflow_t = S_t * f_t$  (5c)

Here, $C$ is the carbon stock in industrial roundwood at the beginning of year $t$ in Mt C. $k$ is the decay constant of first order decomposition for industrial roundwood in $yr^{-1}$. $k$ takes a value of *ln(2) half-life$^{-1}$* of industrial roundwood (half-life assumed to be 35 years here based on IPCC (2019)). *inflow* is the inflow to the non-decayed industrial roundwood pool during year $t$ in Mt C $yr^{-1}$. $\Delta C$ is carbon stock change in the industrial roundwood pool during year $t$ in Mt C $yr^{-1}$. $S$ is the domestically produced industrial roundwood in each region and $f$ is the share of domestic stock for the production of a particular HWP. $f$ values are taken from Johnston and Radeloff (2019). As carbon stored in HWPs is a function of timber demand, it is directly influenced by developments of socioeconomic factors including population, income, and trade akin to timber demand in MAgPIE. Calculation of long-term carbon storage in harvested wood products is documented in Bodirsky et al. (2021).

## 3  Results

### 3.1  Global land-use change

Global land cover and land-use change dynamics over time in the default scenario and the forestry scenario (both SSP2) are shown in Table 2 (rounded to nearest 0) and fig. 10.

In MAgPIE, once natural forests are harvested, the area can be converted to either agricultural land or timber plantations if such expansions are necessary. In the default scenario, we observe that agricultural land (cropland and pasture land) increases by 680 Mha in 1995-2100, mainly at the expense of forests. A smaller increase is seen in the forestry scenario where agricultural land increases by 631 Mha at an expanse of forests as well as other land indicating that cropland intensification takes place when timber production is included. Timber plantation area increases by 132 Mha in forestry scenario to satisfy a considerable portion of industrial roundwood and wood fuel demand from plantations, given the increasing timber demand due to income and population growth. Primary and secondary forest area declines by 405 Mha and 436 Mha respectively between 1995 and 2100 due to the expansion of cropland and timber plantations in the forestry scenario. Other land area decreases by 112 Mha between 1995-2100 in the forestry scenario (as compared to 392 Mha in the default scenario).





**Table 2.** Modeled land-use change between 1995 and 2100 (Mha)

| Land-use | Default | | | Forestry | | |
|---|---|---|---|---|---|---|
| | 1995 | 2100 | 2100-1995 | 1995 | 2100 | 2100-1995 |
| Cropland | 1473 | 2153 | 680 | 1480 | 2111 | 631 |
| Pasture & Rangeland | 3283 | 3554 | 271 | 3288 | 3416 | 128 |
| Forest | 4001 | 3442 | -559 | 4013 | 3366 | -647 |
|    Primary forest | 1366 | 1102 | -264 | 1345 | 940 | -405 |
|    Secondary forest | 2437 | 2079 | -358 | 2461 | 2025 | -436 |
|    Planted forest | 198 | 261 | 63 | 207 | 401 | 194 |
|      Plantations | 84 | 84 | 0 | 93 | 225 | 132 |
|      Afforestation | 114 | 177 | 63 | 114 | 176 | 62 |
| Urban land | 39 | 39 | 0 | 39 | 39 | 0 |
| Other land | 4007 | 3615 | -392 | 3983 | 3871 | -112 |
| Total | 12803 | 12803 | | 12803 | 12803 | |

To satisfy food and feed demand and to accommodate the land-use competition between cropland and forestry, MAgPIE estimates an agricultural yield-shift of 114% and 117% in the default and forestry scenarios respectively by 2100 relative to 1995 through investments in yield-increasing technological change. Such yield-increasing technological change is realized via agricultural land use intensity in MAgPIE and is measured using a $\tau$-factor developed by Dietrich et al. (2012). Global and regional land-use intensity indicator $\tau$ for the forestry and default scenarios is shown in fig. A3.





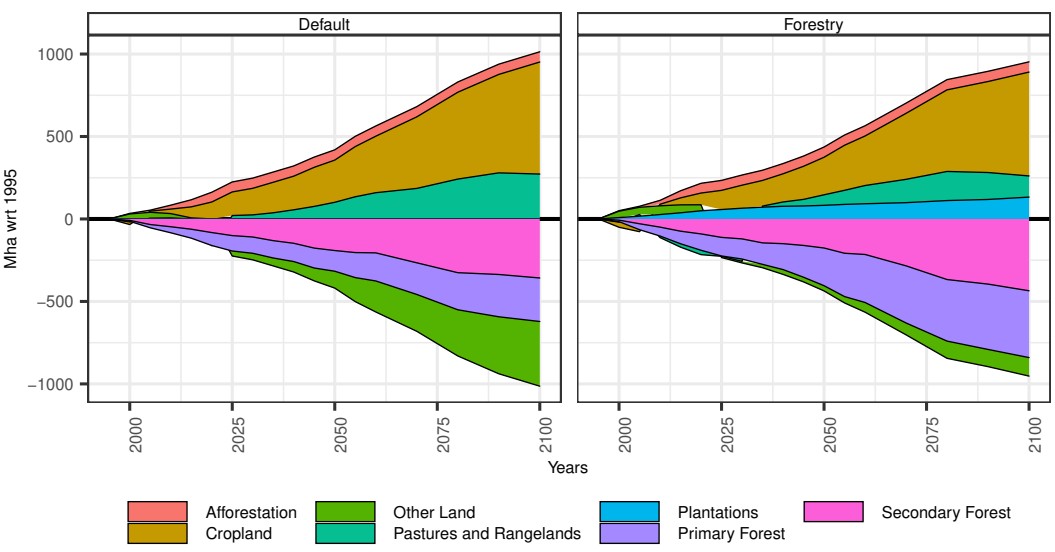

**Figure 10.** Relative land-use change between 1995 and 2100 at global level for default and forestry scenarios. All values wrt 1995 (Mha).

Figure 11 shows the global development and trends in development of plantation area from 1995-2100 (regional development

in fig. A1. Till 2020, MAgPIE matches the historical trend very well, while the levels are slightly higher when compared to the observed data.

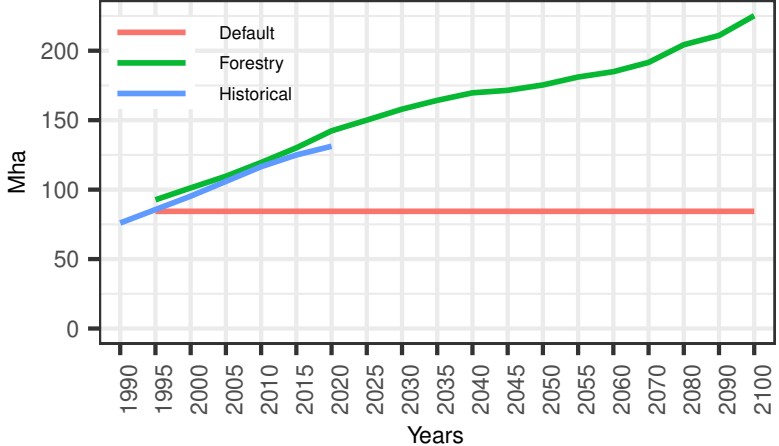

**Figure 11.** Development of plantation area for 1995-2100 at global level in default and forestry scenarios. Flat-line in default scenario is due to the assumption of static plantations at 1995 levels. Historical numbers from Forest Resources Assessment Report (FRA) 2020 (FAO, 2020b).

Default scenario has no changes in plantation area over time due to assumption of static plantations. Figure 12 shows the changes in timber plantation area observed with forestry scenario in 2100 on a 0.5°grid. In absolute terms, the highest gains in





plantation areas are seen in China which will host about 40% of global plantations in 2100 (95 Mha out of 225 Mha). Changes
in natural forest area (primary and secondary forest) in default and forestry scenario is shown in fig. A2.

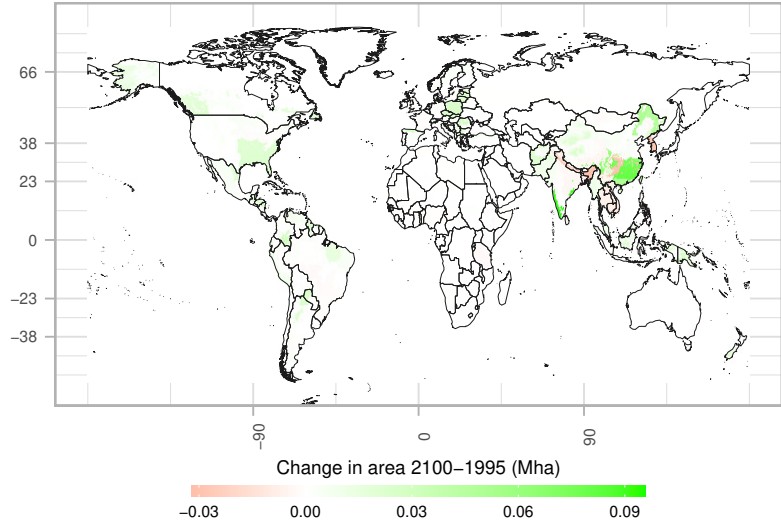

**Figure 12.** Difference in cellular plantations area for forestry scenario between 1995-2100 (Mha).

As plantations compete with cropland for limited land resources, it is important to see how the inclusion of roundwood
production interacts with cropland usage globally. Figure 13 shows the difference in cellular cropland area between forestry
and default scenarios on a 0.5°grid and Table 3 shows the regional differences for the same.

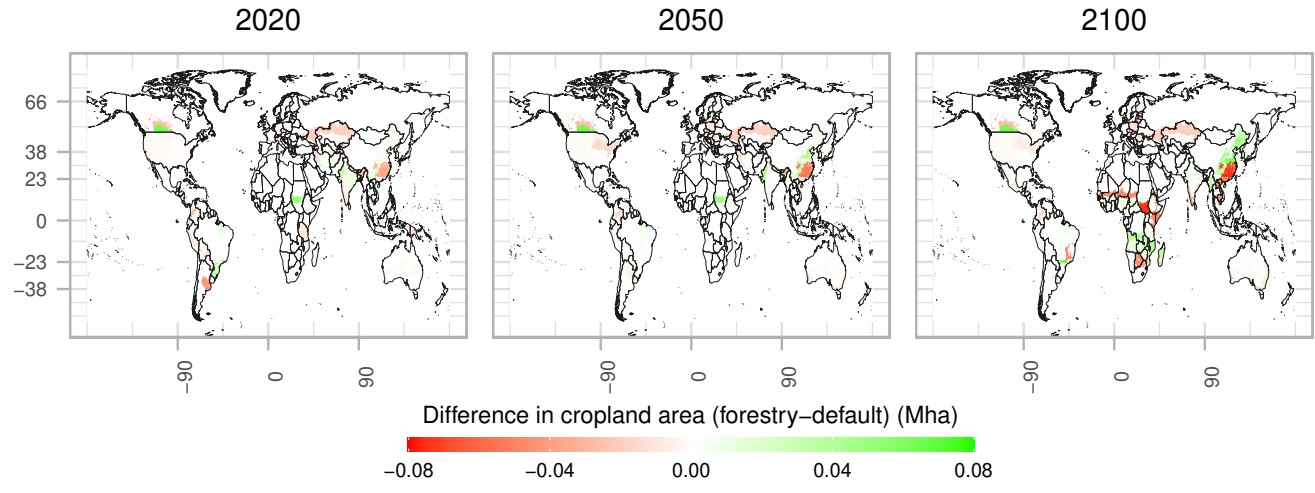

**Figure 13.** Difference in cellular cropland area between forestry scenario and default scenario (Mha) in 2020, 2050 and 2100. Shades of red
indicate cropland loss and shades of green indicate cropland increase when timber production is accounted for in MAgPIE.





**Table 3.** Absolute differences in cropland area (Mha) between forestry and default scenarios.

| MAgPIE regions | 2020 | | | 2050 | | | 2100 | | |
|---|---|---|---|---|---|---|---|---|---|
| | Default | Forestry | Forestry-Default | Default | Forestry | Forestry-Default | Default | Forestry | Forestry-Default |
| CAZ | 92 | 98 | 6 | 107 | 111 | 4 | 114 | 120 | 6 |
| CHA | 119 | 110 | -9 | 123 | 116 | -7 | 94 | 99 | 5 |
| EUR | 125 | 121 | -4 | 127 | 124 | -3 | 132 | 126 | -6 |
| IND | 164 | 169 | 5 | 161 | 165 | 4 | 129 | 129 | 0 |
| JPN | 4 | 4 | 0 | 4 | 4 | 0 | 4 | 4 | 0 |
| LAM | 221 | 218 | -3 | 256 | 261 | 5 | 303 | 304 | 1 |
| MEA | 50 | 58 | 8 | 52 | 59 | 7 | 65 | 66 | 1 |
| NEU | 28 | 28 | 0 | 29 | 30 | 1 | 35 | 35 | 0 |
| OAS | 155 | 155 | 0 | 171 | 172 | 1 | 231 | 232 | 1 |
| REF | 199 | 177 | -22 | 199 | 177 | -22 | 199 | 177 | -22 |
| SSA | 246 | 242 | -4 | 315 | 314 | -1 | 660 | 639 | -21 |
| USA | 174 | 171 | -3 | 185 | 176 | -9 | 186 | 180 | -6 |
| World | 1576 | 1552 | -24 | 1729 | 1708 | -21 | 2153 | 2110 | -43 |

## 3.2 Industrial roundwood production

Figure 14 shows the amount of global industrial roundwood production by source of production. In the forestry scenario we observe plantations providing 375 to 1783 $Mm^3$ $yr^{-1}$ of global industrial roundwood production between 1995-2100 (contribution to overall share in fig. A5). As the plantation area increases over time in the forestry scenario, we see an increasing proportion of industrial roundwood and wood fuel demand being fulfilled by harvesting an increasing amount of available plantations.

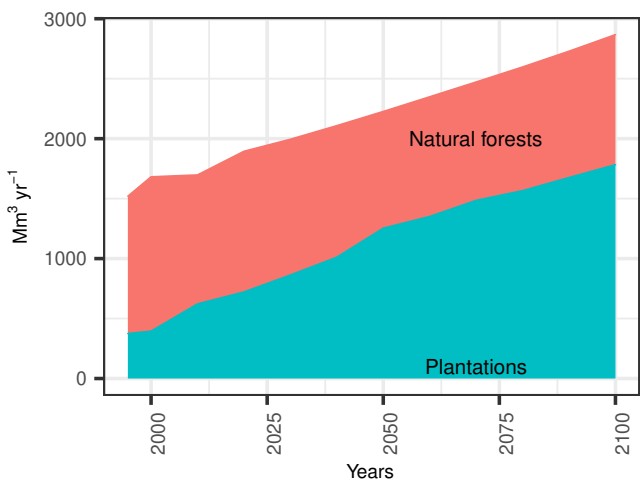

**Figure 14.** Global industrial roundwood production by source for forestry scenario (1995-2100 in Mm$^3$ yr$^{-1}$).

### 3.3 Secondary forest age class structure

Secondary forests are initialized in MAgPIE as described in section 2.4. Once harvested (for timber production) or cleared (for cropland or plantations), secondary forests move to the youngest age class (*ac0*) and are subject to natural regrowth. Primary forests once harvested are re-classified as secondary forest of the youngest age class and follow regrowth. Table 4 shows the difference in secondary forest area between 1995-2100. Development of age class structure in secondary forests for default and forestry scenarios is also shown in fig. 15. Selection of appropriate initial age-class distribution is especially important as they

have a direct relationship with AFOLU emissions (further discussed in section 3.5).

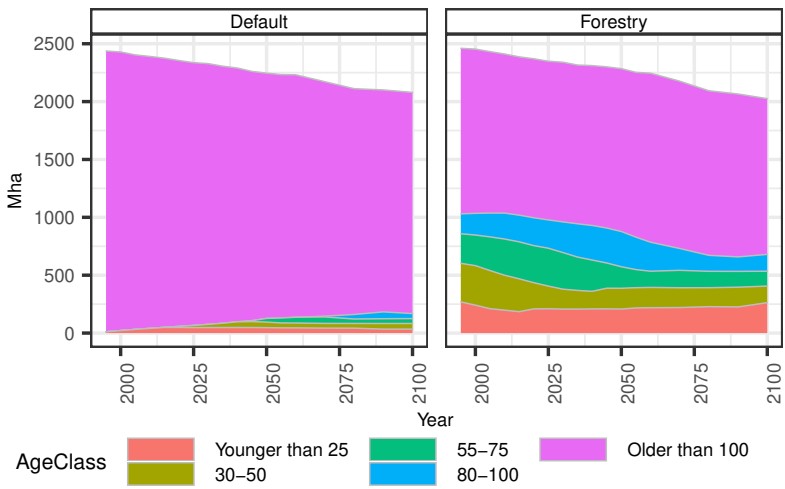

**Figure 15.** Age-class structure in secondary forest. Majority of secondary forest belongs to the highest age class *acx*.





**Table 4.** Difference in secondary forest area 1995-2100 (Mha)

|  | Default | | | Forestry | | |
|---|---|---|---|---|---|---|
| Age-class | 1995 | 2100 | 2100-1995 | 1995 | 2100 | 2100-1995 |
| Younger than 25 | 11 | 34 | 23 | 269 | 263 | -6 |
| 30-50 | 0 | 49 | 49 | 334 | 143 | -191 |
| 55-75 | 0 | 42 | 42 | 255 | 128 | -127 |
| 80-100 | 0 | 43 | 43 | 172 | 145 | -27 |
| Older than 100 | 2426 | 1911 | -515 | 1431 | 1345 | -86 |
| Total | 2437 | 2079 | | 2461 | 2024 | |

## 3.4 Roundwood harvest

Figure 16 shows the annual amount of forest area harvested for meeting the roundwood demand globally (forestry scenario; no harvested area in the default scenario). On average, between 1995-2100, we observe 3 Mha yr$^{-1}$ of plantations and 6 Mha yr$^{-1}$ of natural forest harvest in the forestry scenario. In this scenario, natural forests are harvested more than timber plantations in all periods. In line with the assumptions for timber plantations establishment (increasing share of timber production from plantations in the future), the harvested area from timber plantations increases in the future. Regional details of annual forest area harvested are shown in fig. A7.

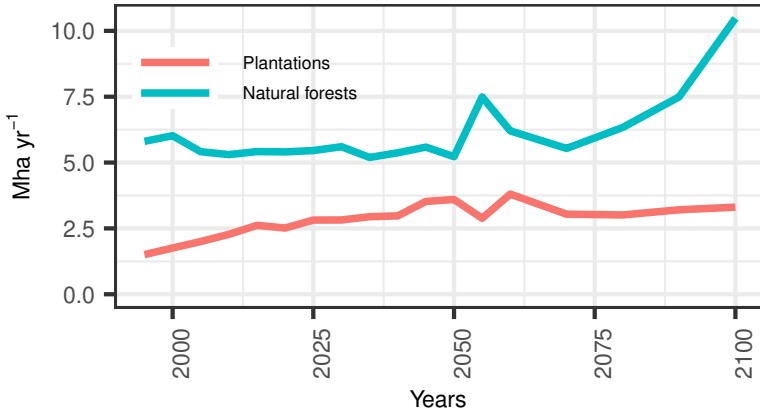

**Figure 16.** Global annual area harvested for roundwood production (Mha yr$^{-1}$) by source in forestry scenario.




## 3.5 Annual Land-use change emissions

Figure 17 shows the annual land-use change emissions from 2000 to 2100. Net Land-use change emission in MAgPIE com-
prises of gross land-use change emissions which include emissions from shifting agriculture (positive), emissions from re-
growth in forests as well as other land (negative) and emissions from wood products (negative, calculated as a net flux between
long term carbon storage in harvested wood products and their slow decay over time).

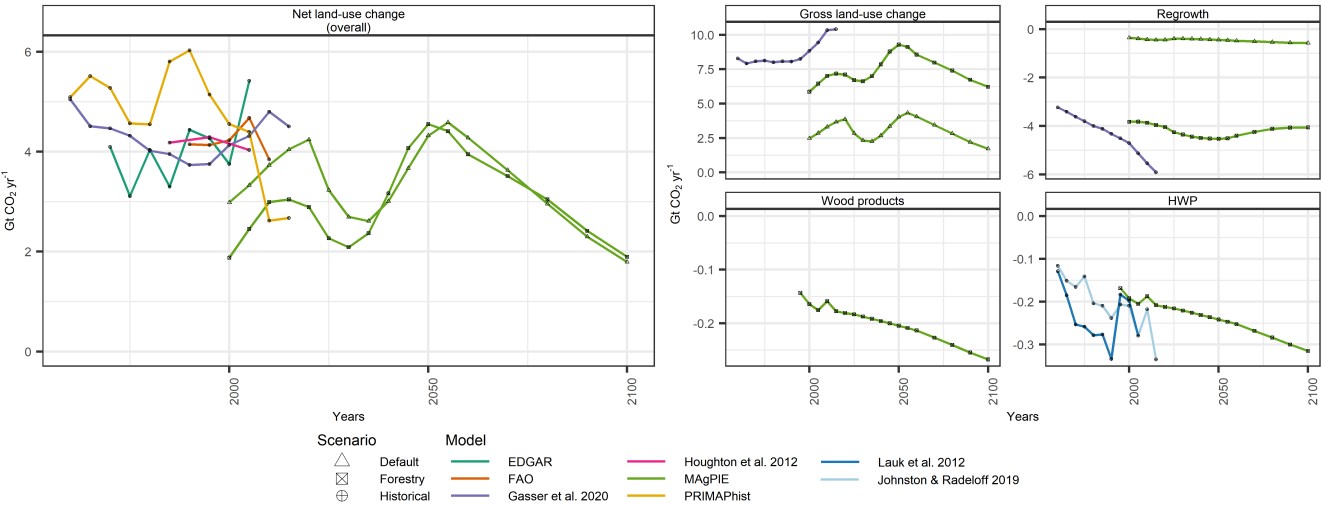

**Figure 17.** Global annual land-use change emissions (Gt $CO_2$ yr$^{-1}$) (1995-2100) and its components. Validation data : Emissions Database
for Global Atmospheric Research (EDGAR) (JRC and PBL, 2010), FAO (2017), Gasser et al. (2020), Houghton et al. (2012), Potsdam
Real-Time Integrated Model for Probabilistic Assessment of Emission Paths (PRIMAPhist) (Gütschow et al., 2016), Lauk et al. (2012) and
Johnston and Radeloff (2019). Regional distribution is available in fig. A6

In the default scenario, land-use change emissions decrease from 3.0 Gt $CO_2$ yr$^{-1}$ in 2000 to 1.8 Gt $CO_2$ yr$^{-1}$ in 2100. In
the forestry scenario we observe that emissions increase from 1.9 Gt $CO_2$ yr$^{-1}$ in 2000 to a peak of 4.5 Gt $CO_2$ yr$^{-1}$ mid-
century and then fall gradually back to 1.9 Gt $CO_2$ yr$^{-1}$ by the end of this century. The net land-use change emissions are
comparable between the default and the forestry scenario. However, the gross land-use change emissions and removals from
regrowth differ substantially between both scenarios. In the forestry scenario, gross land-use change emissions and removals
from regrowth compare much better to value from the literature Gasser et al. (2020). Overall, we represent a historically
consistent representation of regrowth and gross land-use change emissions in the forestry scenario due to accounting for timber
production, and age-class structure in timber plantations and natural forests.

Compared to the default scenario, we observe lower $CO_2$ emissions in the forestry scenario during the initial periods due to
higher carbon uptake driven by assumptions of a heterogeneous initial age-class structure in secondary forests (carbon uptake
can be interpreted as negative emissions where a mathematically lower value is *higher* carbon uptake). In the default scenario,
carbon uptake is much lower because of two reasons : 1) During initialization, all secondary forest is assumed to exist in





the highest age-class, which limits the amount of regrowth, and 2) No secondary forest is harvested for timber production in the default scenario. Without such disturbances, the age-class structure in secondary forests does not shift much towards the younger age-classes (also seen in fig. 15) where usually regrowth is faster as compared to old-age forests.

## 4  Discussion

In this paper, we expanded the MAgPIE modeling framework by a detailed representation of land-use dynamics in natural
forests and timber plantations while accounting for roundwood production and competition for land with agriculture. Representing forestry and timber production in a recursive-dynamic land-use model is a challenging issue due to complexities associated with long term planning horizons needed for roundwood production and forest management. This explains why major land-use models focus on better representation of the agricultural sector or the forestry sector, but not on the competition between both within the same model (Calvin et al., 2019; Wise et al., 2014; Stehfest et al., 2014; Kallio et al., 2004; Havlík
et al., 2011; Kindermann et al., 2006; Sohngen et al., 1999). As timber, food and feed production happen simultaneously in the real world, the inclusion of the forestry sector, next to the agricultural sector, substantially improves the representation of land-dynamics and GHG emissions in MAgPIE.

While including the forestry sector in MAgPIE, we present historically consistent development of timber plantation area over time when compared to observed data (FAO, 2020b). We also present a historically consistent development of growing
stocks in plantations and natural forests over time (FAO, 2020b). Our results show that the inclusion of timber production and plantation establishment in the MAgPIE modeling framework competes with cropland for limited land resources. While the total global cropland is similar between the default and the forestry scenario at the global level, the spatial cropland patterns differ substantially between the two scenarios, which indicates that timber plantations compete with cropland for the same scarce land resources. The net effect is a stronger decline of natural forest in the forestry scenario as compared to the default
scenario. New timber plantations might be partly established on cleared natural forest. However, considering the substantial changes in spatial cropland patterns it seems likely that plantations are also established on agricultural land and pasture land, which causes deforestation for cropland expansion elsewhere.

Our land-related $CO_2$ emissions and removals match better with observed data (Houghton et al., 2012; Gasser et al., 2020; FAO, 2017; Gütschow et al., 2016; JRC and PBL, 2010) in the forestry scenario as compared to the default scenario, in
particular the gross land-use change emissions, reflecting the higher deforestation for expansion of managed land and timber production, and the carbon uptake, reflecting the regrowth in natural forests and timber plantations.

Our modeling study also indicates that timber plantations are an important source of roundwood production. If timber plantations would not increase, in contrast to our forestry scenario, the projected increase in roundwood demand would need to be fulfilled by wood harvest from natural forests. Of particular importance is that plantations can produce more timber
on less area, making them a candidate for reducing roundwood production pressure from natural forests. This opens up a similar question with respect to land-sharing versus land-sparing debate. Establishing high yielding plantations for roundwood





production might provide the benefit of producing a large quantity of timber using a small land area but such plantations do not synergize well with biodiversity. Species richness in plantation forests is usually significantly lower than in natural forests (Phillips et al., 2017). When plantations are established after clearing natural forests, there will be a decline (or even loss) of

biodiversity. On the contrary, it is also important to keep in mind that even when timber plantations embody lower species richness than natural forest in comparable geographic location, plantations if established on degraded land will almost always support higher species richness (Brockerhoff et al., 2008). Plantations may generally be lower in biodiversity, but eventually spare natural forests for $CO_2$ sequestration, biodiversity and soil preservation purposes (Moomaw et al., 2020; Waring et al., 2020; Buotte et al., 2020).

We are aware that our research may have certain limitations as extending a recursive dynamic land-use model to include a dynamic forestry sector is not straightforward and includes some strong generalizations. First, we do not yet account for climate change in this study and our analysis ignores future bio-geophysical changes that come with future climate change. In principal, the modelling framework is capable of accounting for climate change impacts. However, in this study we deliberately chose to focus on the overall forestry implementation and the implications on land-use dynamics and GHG emissions.

Second, the choice of rotation lengths in plantations is an important component for managed forests that follow even-aged management systems. To the best of our knowledge, the determination of optimal rotation lengths for timber plantations has not been done in any of the uncoupled recursive dynamic models so far (Kallio et al., 2004; Calvin et al., 2019; Havlík et al., 2011). The single rotation-period model in MAgPIE does not incorporate the opportunity costs from lost land rent (by ignoring future rotations) resulting in higher rotation lengths when compared to Faustmann rotations. For example, in North

American temperate forest, single rotation period model rotation length (31 years) are 30% longer than Faustmann rotation ages (22 years). In the Scandinavian boreal forest, single rotation period model rotations (60 years) are only 4% longer than Faustmann rotation ages (58 years) (Amacher et al., 2009). Rotation lengths calculated in MAgPIE are not endogenous and are only affected by the prevailing interest rate in the economy but unchanged by fluctuations in timber prices which may not be the correct representation of reality. Using Faustmann rotations in MAgPIE would likely result in somewhat higher

land-use change emissions and lower yields at the time of harvest. Lower yields at harvest would also mean that a larger area of plantations has to be established for meeting the future timber demand, resulting in a higher land demand for plantation establishment, causing additional pressure on the land system.

Third, in forests managed for timber production, thinning is practiced by removing the smaller and poorer quality trees. This operation generates income with the sale of harvested timber and also makes sure that growth is favorable for the remaining

trees. This operation also results in a higher volume and quality of harvested timber, which can generate a higher income in future as the price for such timber is higher in the market. We do not simulate this activity in our updated modeling framework and thereby underestimate the amount of roundwood production capabilities of timber plantations.



## 5    Conclusions

Since the inception of MAgPIE, the modeling framework has evolved with time to include a broad range of land-use processes.
In this paper, we describe an extension of the existing MAgPIE framework by a detailed representation of timber demand and
production, forest land and timber plantations. MAgPIE 4.3.2 allows land-use processes for timber production to be simulated
with feed, food and livestock demand simultaneously, advancing the land-use representation from previous MAgPIE versions.
Given the growing importance of timber plantations in meeting growing global timber demand, it is also imperative that timber
plantation systems are modeled explicitly. within forest systems in land-use modeling. Timber production has not been a part
of the MAgPIE modeling framework since its inception, which means that a major driver for deforestation and land-use change
emissions has been missing. With this paper, we bridge this gap and expand the coverage in the representation of most relevant
land-use change drivers in MAgPIE.

Inclusion of the forestry sector in MAgPIE offers improved understanding of land resources, which plays a vital role in
climate change mitigation (Doelman et al., 2018), biodiversity conservation (Gibson et al., 2011; Phillips et al., 2017) and
maintaining crucial ecosystem services (Foley et al., 2005). This expanded version of MAgPIE not only provides an improved
tool for comprehensive assessments of the Sustainable Development Goals (SDG) but may also contribute to other important
scientific processes, such as providing inputs for Earth System Models (ESMs) (Hurtt et al., 2018; Luyssaert et al., 2014;
Reid et al., 2010; Bonan and Doney, 2018), Biodiversity models (Thuiller et al., 2013; Urban et al., 2016), or international
networks like the Agricultural Model Inter-comparison and Improvement Project (AgMIP) (Ruane and Rosenzweig, 2018) or
the Inter-Sectoral Impact Model Inter-comparison Project (ISIMIP, www.isimip.org).





*Code and data availability.* The MAgPIE code is available under the GNU Affero General Public License as published by the Free Software Foundation, version 3 of the License or later (AGPLv3) via GitHub (https://github.com/magpiemodel/magpie, last access: 04 March 2021). MAgPIE release version (v4.3.1) on which this paper is built-on can be found via Zenodo https://doi.org/10.5281/zenodo.4231467 Dietrich et al. (2020b). The technical model documentation is available under https://rse.pik-potsdam.de/doc/magpie/4.3/ (last access: 04 March 2021) and archived via Zenodo (https://doi.org/10.5281/zenodo.1418752). MAgPIE model results shown in this paper (including model code) are archived via Zenodo (https://doi.org/10.5281/zenodo.4576400). Model code used in this paper is also available via https://github.com/abhimishr/magpie/releases/tag/v4.3.2 on GitHub (additional link). This will be replaced by an official MAgPIE release later.

## Appendix A

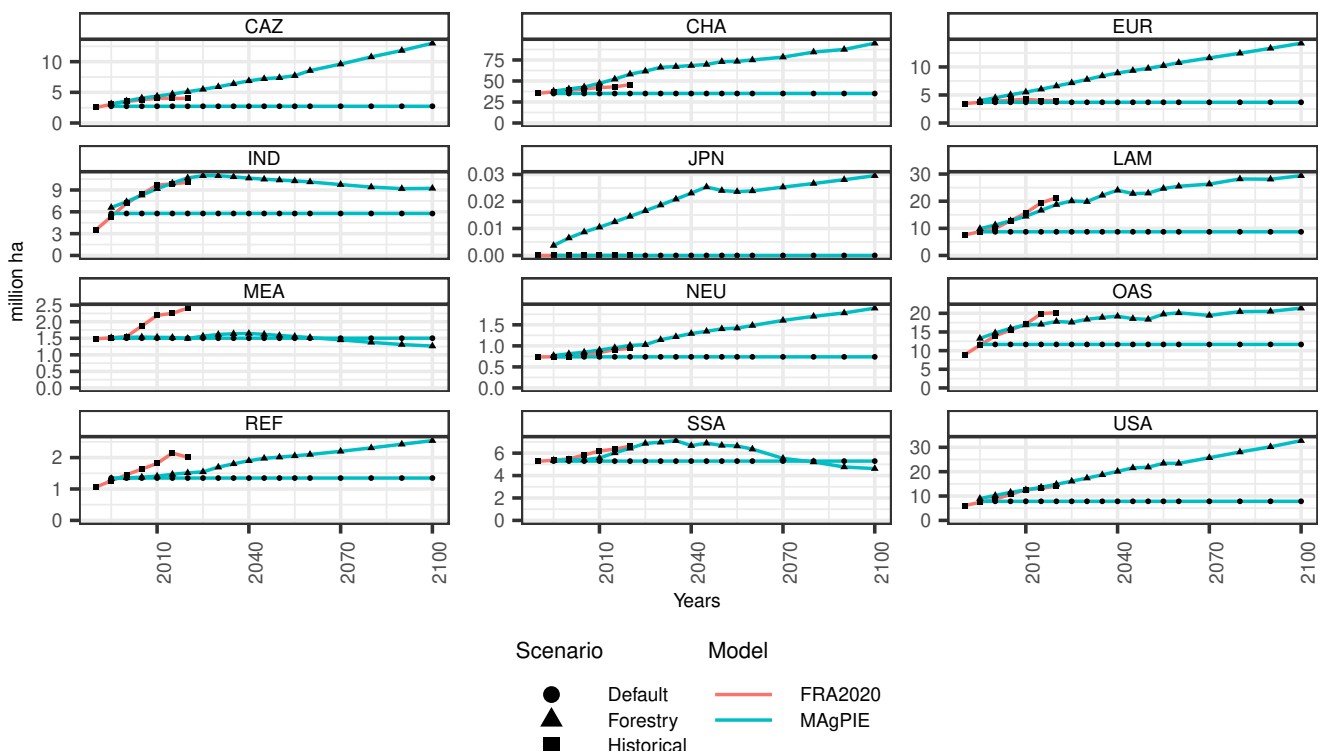

**Figure A1.** Regional development of plantation area for 1995-2100 in SSP2 scenario.



**Figure A2.** Natural forest area difference between 2100-1995 in default and forestry scenarios.



**Figure A3.** a) Global and b)Regional Land-use Intensity Indicator (TAU) as a productivity measure (Index)



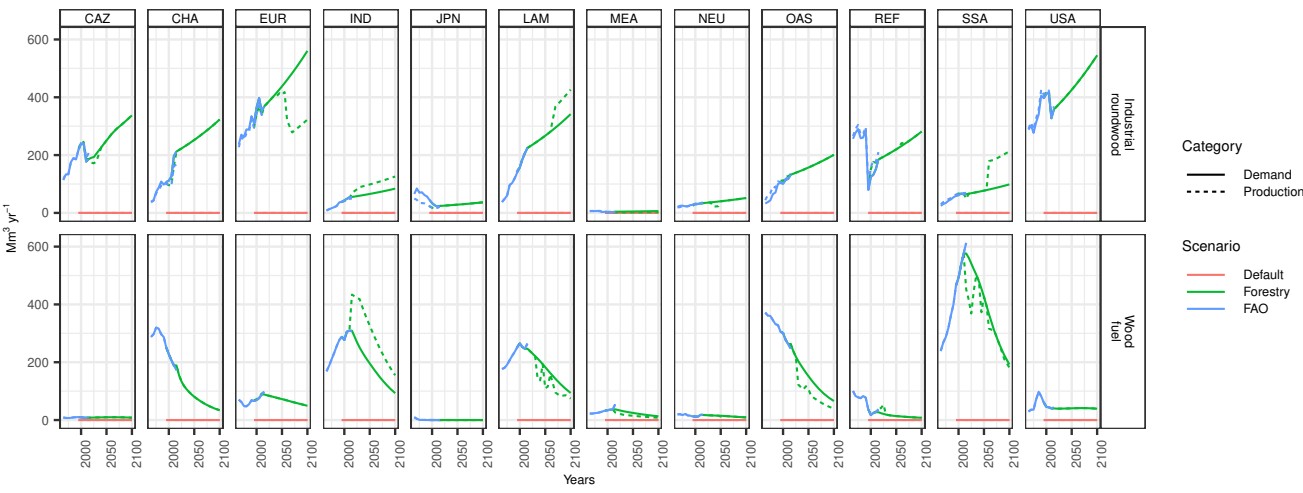

**Figure A4.** Production and demand of industrial roundwood and wood fuel in $Mm^3 \ yr^{-1}$.

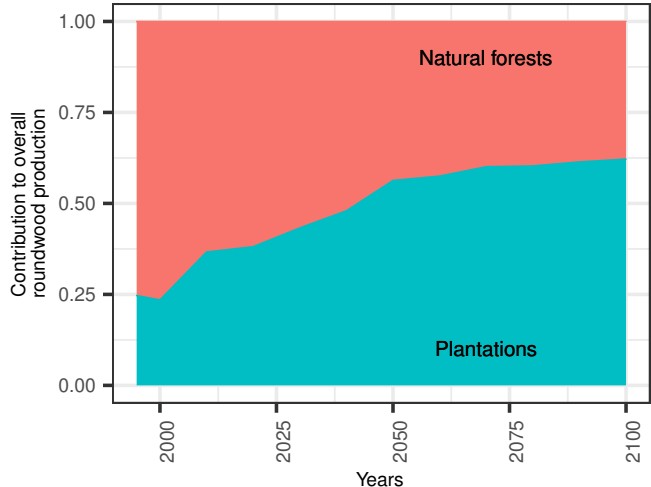

**Figure A5.** Contribution of timber harvest from natural forests and plantations to industrial roundwood and wood fuel production in forestry scenario (1995-2100).



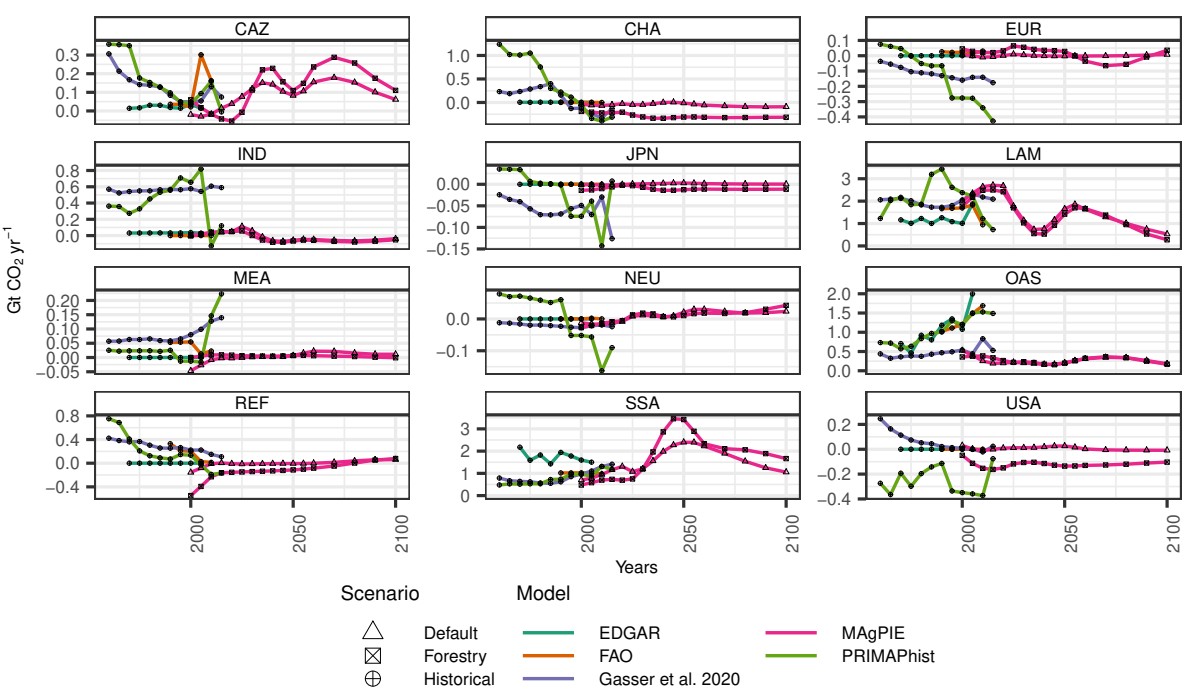

**Figure A6.** Regional annual net land-use change emissions (Gt $CO_2$ yr$^{-1}$) (1995-2100).

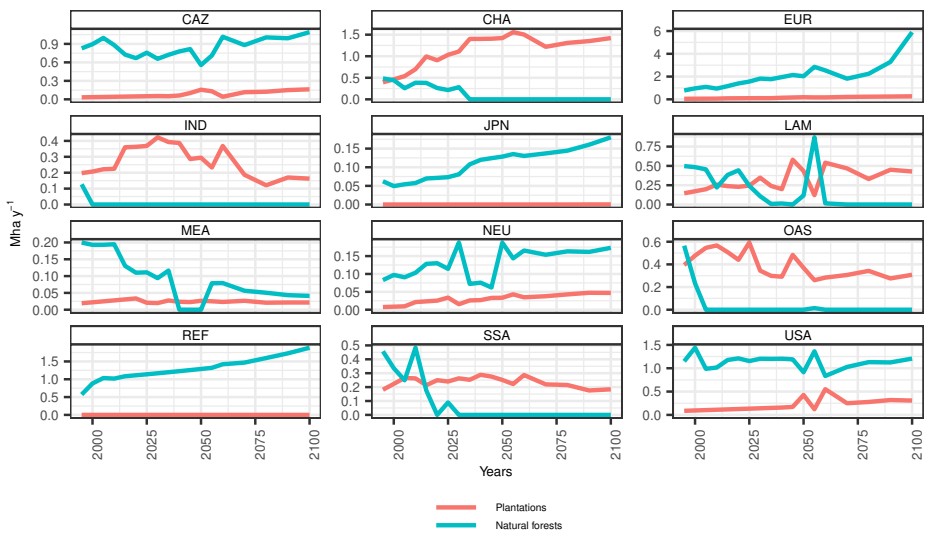

**Figure A7.** Regional annual area harvested for roundwood production (Mha yr$^{-1}$) by source.



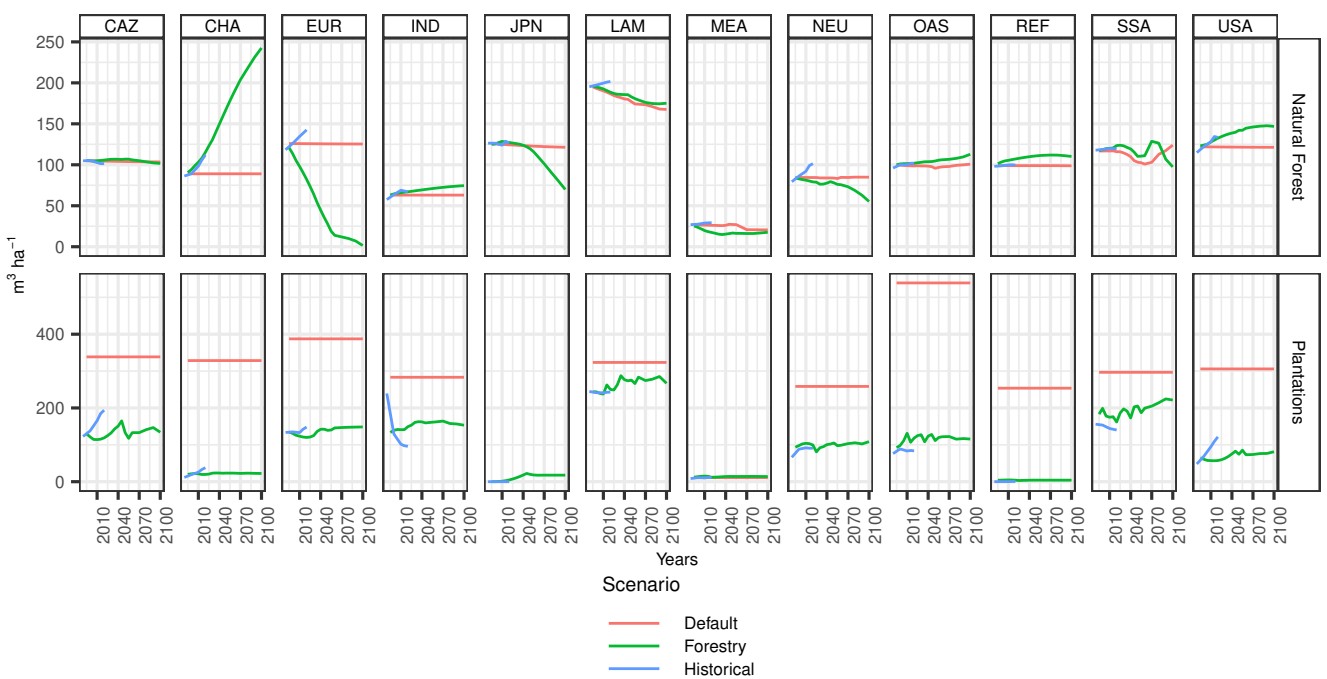

**Figure A8.** Regional growing stocks in natural forests and plantations (m³ ha⁻¹).





**Table A1.** ISO3 codes of countries belonging to standard MAgPIE regions.

| MAgPIE Regions | ISO3 country codes |
|---|---|
| CAZ | AUS; CAN; HMD; NZL; SPM |
| CHA | CHN; HKG; MAC; TWN |
| EUR | ALA; AUT; BEL; BGR; CYP; CZE; DEU; DNK; ESP; EST; FIN; FRA; FRO; GBR; GGY; GIB; GRC; HRV; HUN; IMN; IRL; ITA; JEY; LTU; LUX; LVA; MLT; NLD; POL; PRT; ROU; SVK; SVN; SWE |
| IND | IND |
| JPN | JPN |
| LAM | ABW; AIA; ARG; ATA; ATG; BES; BHS; BLM; BLZ; BMU; BOL; BRA; BRB; BVT; CHL; COL; CRI; CUB; CUW; CYM; DMA; DOM; ECU; FLK; GLP; GRD; GTM; GUF; GUY; HND; HTI; JAM; KNA; LCA; MAF; MEX; MSR; MTQ; NIC; PAN; PER; PRI; PRY; SGS; SLV; SUR; SXM; TCA; TTO; URY; VCT; VEN; VGB; VIR |
| MEA | ARE; BHR; DZA; EGY; ESH; IRN; IRQ; ISR; JOR; KWT; LBN; LBY; MAR; OMN; PSE; QAT; SAU; SDN; SYR; TUN; YEM |
| NEU | ALB; AND; BIH; CHE; GRL; ISL; LIE; MCO; MKD; MNE; NOR; SJM; SMR; SRB; TUR; VAT |
| OAS | AFG; ASM; ATF; BGD; BRN; BTN; CCK; COK; CXR; FJI; FSM; GUM; IDN; IOT; KHM; KIR; KOR; LAO; LKA; MDV; MHL; MMR; MNG; MNP; MYS; NCL; NFK; NIU; NPL; NRU; PAK; PCN; PHL; PLW; PNG; PRK; PYF; SGP; SLB; THA; TKL; TLS; TON; TUV; UMI; VNM; VUT; WLF; WSM |
| REF | ARM; AZE; BLR; GEO; KAZ; KGZ; MDA; RUS; TJK; TKM; UKR; UZB |
| SSA | AGO; BDI; BEN; BFA; BWA; CAF; CIV; CMR; COD; COG; COM; CPV; DJI; ERI; ETH; GAB; GHA; GIN; GMB; GNB; GNQ; KEN; LBR; LSO; MDG; MLI; MOZ; MRT; MUS; MWI; MYT; NAM; NER; NGA; REU; RWA; SEN; SHN; SLE; SOM; SSD; STP; SWZ; SYC; TCD; TGO; TZA; UGA; ZAF; ZMB; ZWE |
| USA | USA |





**Table A2.** Interest rates used in MAgPIE for determination of rotation lengths in plantations.

| MAgPIE region | Interest rate (%) |
| --- | --- |
| CAZ | 0.040 |
| CHA | 0.100 |
| EUR | 0.052 |
| IND | 0.100 |
| JPN | 0.060 |
| LAM | 0.081 |
| MEA | 0.087 |
| NEU | 0.075 |
| OAS | 0.099 |
| REF | 0.073 |
| SSA | 0.097 |
| USA | 0.040 |

**Table A3.** Self sufficiency ratios in MAgPIE for Industrial roundwood and wood fuel for 1995, 2020, 2050 and 2100.

| MAgPIE region | 1995 | | 2020 | | 2050 | | 2100 | |
| --- | --- | --- | --- | --- | --- | --- | --- | --- |
| | Industrial round-wood | wood fuel | Industrial round-wood | wood fuel | Industrial round-wood | wood fuel | Industrial round-wood | wood fuel |
| LAM | 1.04 | 1.00 | 1.00 | 1.00 | 1.00 | 1.00 | 1.00 | 1.00 |
| OAS | 1.02 | 1.00 | 1.05 | 1.00 | 1.05 | 1.00 | 1.05 | 1.00 |
| SSA | 1.07 | 1.00 | 1.06 | 1.00 | 1.06 | 1.00 | 1.06 | 1.00 |
| EUR | 0.95 | 1.01 | 0.96 | 1.01 | 0.96 | 1.01 | 0.96 | 1.01 |
| NEU | 0.88 | 1.00 | 0.97 | 1.01 | 0.97 | 1.01 | 0.97 | 1.01 |
| MEA | 0.77 | 1.00 | 0.73 | 1.00 | 0.73 | 1.00 | 0.73 | 1.00 |
| REF | 1.22 | 1.00 | 1.17 | 1.03 | 1.17 | 1.03 | 1.17 | 1.03 |
| CAZ | 1.00 | 1.01 | 1.06 | 0.99 | 1.06 | 0.99 | 1.06 | 0.99 |
| CHA | 0.95 | 1.00 | 0.82 | 1.00 | 0.82 | 1.00 | 0.82 | 1.00 |
| IND | 0.99 | 1.00 | 0.90 | 1.00 | 0.90 | 1.00 | 0.90 | 1.00 |
| JPN | 0.51 | 1.00 | 0.79 | 1.00 | 0.79 | 1.00 | 0.79 | 1.00 |
| USA | 1.03 | 1.00 | 1.03 | 1.00 | 1.03 | 1.00 | 1.03 | 1.00 |





**Table A4.** Percentage of production which can possibly come from plantations based on Pöyry (1999)

| Region | 1995 | 2020 | 2050 | 2100 |
|--------|------|------|------|------|
| LAM | 0.54 | 0.69 | 0.73 | 0.77 |
| OAS | 0.33 | 0.42 | 0.44 | 0.46 |
| SSA | 0.20 | 0.26 | 0.27 | 0.29 |
| EUR | 0.46 | 0.59 | 0.62 | 0.66 |
| NEU | 0.46 | 0.59 | 0.62 | 0.66 |
| MEA | 0.21 | 0.27 | 0.28 | 0.30 |
| REF | 0.46 | 0.59 | 0.62 | 0.66 |
| CAZ | 0.28 | 0.36 | 0.38 | 0.40 |
| CHA | 0.32 | 0.41 | 0.43 | 0.46 |
| IND | 0.32 | 0.41 | 0.43 | 0.46 |
| JPN | 0.32 | 0.41 | 0.43 | 0.46 |
| USA | 0.22 | 0.28 | 0.30 | 0.31 |

**Table A5.** Calibration factor for establishment decisions

| MAgPIE Region | Calibration factor |
|---------------|--------------------|
| LAM | 2.0 |
| OAS | 1.5 |
| SSA | 1.0 |
| EUR | 1.00 |
| NEU | 1.0 |
| MEA | 0.3 |
| REF | 3.0 |
| CAZ | 1.0 |
| CHA | 1.0 |
| IND | 1.5 |
| JPN | 1.0 |
| USA | 1.0 |



*Author contributions.* AM, FH and AP proposed and led this study. AM, FH and BB wrote the original model extension for forestry and natural vegetation and timber modules. AM, FH and JPD expanded the implementation of drivers, demand, trade and carbon modules. FH, AP, JPD, BB, CR, BS and HLC guided the model development. AM prepared the model input data. FH, JPD and BB provided technical support for the development. FH, CR, JPD and BB provided theoretical support for the development. AM made the model runs and processed the model outputs and produced the figures. AM and FH wrote the additional model documentation. AM, FH, JPD prepared the extended model for release. All authors contributed to the writing and editing processes.

*Competing interests.* The authors declare that they have no conflict of interest.

*Acknowledgements.* The authors thank FAOSTAT, World Bank and the SSP scenario modelers for the data provided which acts as major model drivers. We thank Dr. Benjamin Poulter (NASA), Kristine Karstens (PIK/HU Berlin, Germany), Felicitas Dorothea Beier (PIK/HU Berlin), Dr. Jens Heinke (PIK), Dr. Jonathan Doelman (PBL), Dr. Thomas Gasser (IIASA), Dr. Niklas Forsell (IIASA), Dr. Pekka Lauri (IIASA) and other colleagues at PIK for valuable discussions during the development of the modeling framework. The authors are also grateful for the constant support of IT team managing the High-Performance Cluster (HPC) computers for scientific calculations at PIK.

We also acknowledge Leibniz Association's Economic Growth Impacts of Climate Change (ENGAGE) project under grant no. SAW-2016-PIK-1, Bundesministerium für Bildung und Forschung (BMBF) funded Pathways and Entry Points to Limit Global Warming to 1.5°C (PEP1p5) project under grant no. 01LS1610A, BMBF (DE), BMWFW (AT), NWO (NL), FORMAS (SE) and European Union funded project SENSES under grant no. 01LS1712A as well as BMBF funded Deep Transformation Scenarios for Informing the Climate Policy Discourse (DIPOL) project under grant no. 01LA1809A which funded the research work of Abhijeet Mishra.





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
