# Peer review of "Estimating global land system impacts of timber plantations using MAgPIE 4.3.5"

_Geoscientific Model Development, 2021_

## Author Comment (AC1)

We would like to thank Dr. Pekka Lauri for the time spent on reviewing our paper and the valuable remarks which pointed out some important issues which will help to further improve the paper. Dr. Pekka Lauri's comments are in grey, and our responses are in black with proposed edits for the revised manuscript in italics.

**General comments:**

p12L200-215: The expansion of timber plantation depends on the share of production that comes from plantations ($\eta$). This parameter is exogenous and extrapolated from Pöyry (1999). This means that the expansion of plantation is not endogenous in the model, but it is taken as given. According to figure A5, the model assumes that share plantation increases on average from 25% in 2000 to 62.5 % in 2100. This issue should be made clear already in the abstract because it has significant impact on the results. For example, if the share of plantations were endogenous in the model, then an increasing demand for roundwood would increase the share of plantations relative to natural forests in EUR region. But because the share is exogenous this does not happen, and EUR region is not able adapt higher demand by intensifying their forest management (Figure A6).

Yes, you are right, this is an important point. We will make this limitation clear in the abstract and add a sentence that highlights the assumption regarding the $\eta$ parameter. p1L4 will be changed as follows: (...) elsewhere. *Using exogenous extrapolation of historical roundwood production from plantations and timber demand, we prescribe expansion of forest plantations at the regional level.* As a (...)

The outcome of the optimal rotation models depends much on interest rates and usually these models include sensitivity analysis relative to different interest rate. To avoid this complication, the rotation times could be solved by maximizing increment (f'=f(ac)/ac) instead of maximizing NPV (f'(ac)/f(ac)=r). This would also be more reasonable objective for the recursive dynamic model where all other choices are based on recursive optimization instead of intertemporal optimization.

Thank you for this important comment. We will add this capability (and switch to this way of rotation length calculation) in the model to calculate rotation lengths via maximization of increment. This will also be reflected in the main text and associated equations and figures (eq. 1, fig. 3). The resulting changes to rotation lengths will also be shown in updated fig. 4.

Add some discussion about the forest age-class dynamics and optimal rotation models in the introduction. Basically move some material from discussion to introduction. Including forest age-class dynamics in the large-scale land-use model is the main contribution of the study, but this issue is completely ignored in the introduction.

We will move the segment discussing the forest age-class dynamics and rotation lengths from p1L370:382 to the introduction part.

**Specific comments:**

p2L26: According to FAOSTAT global roundwood demand was 3969 Mm3 in 2019 and industrial roundwood 2024 Mm3. Global roundwood demand cannot be 1683 Mm3.

We will replace this instance of *roundwood* with *industrial roundwood* in p2L26. As there are no new estimates of the share of industrial roundwood production coming from plantations, we assume that the trends observed by Jürgensen et al. 2014 still hold. We also updated the industrial roundwood production data now based on latest FAOSTAT numbers. We will reformulate p2L26 as: (...) *likely supply more than 33% (654 Mm3) of global industrial roundwood demand (1984 Mm3) in 2020 based on historical trends (Jürgensen et al., 2014).*

**p2L26 Add reference or explanation for 33% (560 Mm3) plantation supply -> Pöyry (1999) extrapolation (Figure A5). This it is not data but model outcome.**

We will add Jürgensen et al. 2014 as a reference for this sentence. In fig. A5, an additional sentence will be added to the caption with a new formulation as: *Contribution of timber harvest from natural forests and plantations to industrial roundwood and wood fuel production in the forestry scenario (1995-2100) based on extrapolations from Pöyry (1999).*

New reference: Jürgensen, C., Kollert, W., & Lebedys, A. (2014). Assessment of industrial roundwood production from planted forests. *Planted Forests and Trees Working Papers (FAO) eng no. FP/48/E.*

**p2L38: Add more relevant references for high roundwood productivity of plantations relative to natural forests than FAO (2013), e.g. IPCC (2006). Also, add some explanation why roundwood productivity is higher in plantations than managed natural forests.**

We will add the following references for high roundwood productivity of plantations relative to natural forests:

a) IPCC 2006
b) Payn et al. 2015
c) Cubbage et al. 2007
d) Evans and Turnbull 2004

From a qualitative point of view, plantations have more control on breeding material, fertilization, management intensity etc. than managed natural forests and hence more control of quality and quantity. We will also add the following explanation for higher productivity in plantations than managed natural forests in p2L37: (..)imperative. P*lantation forests for timber production have potentially higher annual average increment per area than natural forests and managed natural forests (IPCC 2006, Payn et al. 2015, Cubbage et al. 2007, Evans et al. 2004) because they are managed more intensively (fertilizer, thinning) and rely on high quality seeds and seedlings for regeneration. Because of* their (...)

**p6L124, p6L132, p7L136: Equation should be f '/ f=r.**

We will correct the formulation in eq. 1. The changes requested in general comments – no.2 will also be reflected in this proposed change to eq. 1:

$$f'_{ac} = \frac{f_{ac}}{ac}$$

**Rotation times for timber plantations in Figure 4 are "interesting", but the question is how reasonable they are. For example, with 30-40 years rotation time in Russia and**

Europe you get only pulpwood (sawlogs require 60-100 years rotation). Moreover, it is not clear why rotation times are longer in North-America than in Europe and Russia. Is this connected to interest rates or productivity? There is only a small difference in interest rates (Table A2) and there should not be large differences in biomass growth between these regions. Some discussion of this should be added and eventually an update to growth curves, interest rate data and add a minimum diameter constraint for sawlogs.

This is a very important point. We noticed that not having enough heterogeneity in the parametrization of our growth function with underlying parameters from Braakhekke et al. 2019 resulted in relatively homogenous rotation lengths within MAgPIE regions. We will change our rotation length calculation to maximize increment as suggested in general comments no.2 – decoupling the calculation from dependence on interest rate. We will also add some additional explanations regarding the assumptions for growth curves. In MAgPIE we do not use or model the minimum diameter constraint for sawlogs. Biomass extraction from trees is calculated based on expected yield and area information for simplicity.

Is rotation time for natural forests determined by the same rule than for timber plantations (equation 1). If yes, then add similar map (Figure 3) for natural forest rotation time. It would be interesting to see the regional difference between timber plantations and natural forests rotation times. If no, then add some justification why natural forest rotation time is chosen differently than in timber plantations. Basically explain also natural forest rotation lengths in chapter 2.3.

Natural forests are not bounded by rotation length constraints. The model is free to choose which age-class in natural forests to harvest based on harvesting costs and associated trade-offs i.e., during each optimization step, while harvesting natural forests, a decision is made whether it would be cheaper to harvest from alternative sources i.e., plantations. MAgPIE's objective function is to minimize global production costs. We use a lower harvesting cost (per ha) for plantations than in natural forests. This implicitly provides a signal to the model to harvest forests with higher growing stock first. We will add an additional sentence in p12L221 for clarity: (...) and maturity *as natural forests are not bounded by rotational constraints*.

We will also rename section 2.3 to *Rotation lengths* instead.

According to Figure A8 EUR region growing stock decreases close to zero in 2100, which implies that forest management is not sustainable in EUR region. Easy way to avoid this would be to add additional "sustainability" constraint on harvests (harvests ≤ α x increment where α=1 for normal forests, α> 1 for old forests and α < 1 for younger forests). Alternatively, increase the share of plantation in EUR region. Basically take account in extrapolation of η that demand increases in the future. Interesting is also opposite development of growing stock in CHA region. This seems to be connected to the higher share of roundwood production coming from plantation in CHA region (Figure A7). Why cannot EUR region do the same as CHA region and avoid the decrease in growing stock ?

We will increase the share of production which can possibly come from plantations in EUR. We can also confirm (based on a new model run with proposed changes) that altering rotation length calculations by maximizing increment as proposed in general comments no.2 and increasing the share of production than can in principle come from plantations in EUR (0.54 in 1995 to 0.86 in 2100) will result in stable growing stock development over time in EUR.

Updated numbers will be shown in table A4 and updated results will be shown in fig. A5, fig. A7 and fig. A8 in the revised manuscript.

---

## Author Comment (AC2)

We would like to thank Mr. Walter Rossi Cervi for the time spent on reviewing our paper and the valuable remarks which pointed out some important issues which will help to further improve the paper. Mr. Walter Rossi Cervi's comments are in grey, and our responses are in black with proposed edits for the revised manuscript in italics.

**General comments:**

I understood the principle of rotation lenghts derived from the relation between interest rate and IGR. This generates information on rotation lenghts and it is important for calculating timber production costs. I was wondering whether some reference data could be added to validate the rotation lenght map. I`m not a forest specialist, but in principle Latin America and Australia should have the similar spatial pattern, given that that both regions are important eucalyptus producers. Instead, the map shows different spatial patterns. I believe that an analysis on the main species (and they respective rotation period) per region could be used to calibrate the rotation lenght map.

Thank you for this important comment. Using FAO 2006 (Global planted forests thematic study. Results and Analysis), the mean rotation length in Australia is 34 years (min = 28 years, max = 40 years) - mean rotation length in Latin America (using South America from FAO reported numbers) is 24 years (min = 18 years, max = 29 years). Leech 2013 suggest an optimal rotation of 45 years in Australia. Additionally, the intensity of the management, e.g. using genetic engineered plant material, irrigation and fertilization might differ between Latin America and Australia a lot, even for the same species. As these are aggregated numbers, it cannot be determined with certainty if the spatial patterns on a finer spatial scale (as in fig. 4) are going to be similar between Latin America and Australia. In principle, the numbers from FAO 2006 can be used to validate the rotation lengths from MAgPIE and we can add this as an additional figure in the manuscript.

MAgPIE in its current format cannot handle tree species information. Calculation of rotation lengths at the cellular level is one of the novelties of this manuscript. Calibrating of cellular rotation lengths to a regional or country level data-set would result in loss of the spatial (cellular) level differentiation in rotation lengths. Instead, for the initial time step, we calibrate the growing stocks to FRA 2020 reported numbers for both natural forests and plantations. Also using a single value for rotation length per MAgPIE region would not be ideal as there are spatial differences in the way plantations grow within each region. Our way of deriving rotation lengths based on carbon stock information from LPJmL for natural vegetation (different species dominate in different cells) helps us in using spatial differences in carbon densities to act as a proxy for differences in species. We are also not aware of any spatially explicit data which is on a finer spatial scale (e.g., 0.5° resolution) with information on both tree species and associated rotation length which can be used to calibrate MAgPIE's cellular level rotation lengths correctly.

It is clear that the study was focused on creating a proof of concept that enables a forestry module in MAGPIE. However, I missed a bit the discussion on how realistic are the figures presented. For example, in section 3.1, it is mentioned a large increase in cropland at expense of primary forest areas (where exactly?). Are protected areas included in the analysis? What about agro-forestry (also included?)? Perhaps some discussions around the current and future spatial uncertainties would be relevant as well.

We will add further discussion on how realistic the figures are based on uncertainties in the socio-economic model drivers across SSPs as this manuscript only considers the SSP2 scenario.

We will also add a regional level figure akin to fig. 10 in the appendix to clarify where does cropland expand at the cost of primary forests (also including other land-use types).

In terms of protected areas, the manuscript accounts for National Policies Implemented (NPIs) in terms of forest protection and afforestation according to existing national policies until 2030, in support of the Paris Agreement. Additional land protection is based on The World Database on Protected Areas (WDPA) which earmarks category I and II areas from International Union for Conservation of Nature (IUCN) as protected in MAgPIE. We will make this further clear in the method and discussion section.

Agroforestry is not included in MAgPIE.

We will add further discussion on the uncertainty of spatially explicit data on plantation forest, with respect to the differentiation between productive and non-productive plantations – which in turn also has a bearing on the results in this manuscript and that the management of plantations also depends on other factors such as availability of workforce, investment, R&D available to improve the management etc.

Productivity is an important component for calculating global costs of demand driven land uses. I`m not fully aware of  LPJML, but doing a quick research, I found that LPJML might include the productivity of natural vegetation and planted forests. These were surely incoporated in MAGPIE (right?). But yet it is not clear to me how the productivity of secondary forest came about?

For every cell, the long-term carbon density of natural vegetation from LPJmL is converted to age-class dependent carbon densities by a Chapman-Richards volume growth function. Based on information from LPJmL, we use different parameters in the Chapman-Richards function for plantations and natural vegetation, resulting in faster regrowth of plantations as compared to natural vegetation. The resulting age-class dependent carbon densities for plantations and natural vegetation are converted to harvestable yield with the help of biomass conversion and expansion factors. For secondary forests, we use these age-class specific harvestable yields. We initialize the age-class structure of secondary forest based on observational data (Poulter et al). For primary forests, we use the highest age-class yield, which reflects natural vegetation. This is also explained in eq. 3 and p10L189, where age-class specific natural vegetation yields are akin to secondary forests. We will further clarify it with an additional sentence.

We will further clarify it with an additional sentence in p10L190: (...) tDM/ha, *yr is age-class and forest type specific harvestable yield*, C is the (...).

To avoid confusion, we will add a sentence in the introduction of the manuscript which makes it clear that we include secondary forest in our definition of the term "natural forest", based on the rationale that secondary forest is regrown natural forest. We will change p10L193 as: (...) et al. (2020a). *Harvestable biomass yield (yr) is different between natural forests (primary and secondary forests) and plantations by virtue of differences in the parametrization of underlying growth function(s).*

**Specific comments:**

Line 175. If a fraction of forestry residues is recovered during the harvesting period, it is likely that there will be a potential decay in productivity in the forthcoming period in comparison with a plantation system that does not recover any fraction of residues. It was not cleat for me if that was included in the model, but it is something to be considered

We assume that the residues are collected from the overall production system i.e., we do not differentiate if the residue comes after harvest from plantations or natural forests. Additionally, the decay in productivity after residue removal is also not modelled. The residues are used to fulfil a part of wood fuel demand as described in p9L175:178. This residue generation constraint in the model is an upper bound and the model is flexible to decide based on the cost of production if the residue should be removed or not from the part of production which comes from plantations. We will add a short discussion on it in the revised manuscript, making it clear that in MAgPIE, only a fraction of the residues are removed depending on production costs but if that happens, no decline in nutritional status is assumed, which is a caveat. We would also argue that at least for some plantations fertilization would be applied to maintain productivity.

**Typo:**

Line 119. "optimal"

The line containing this typo will be removed as we will re-write the rotation length calculation segment as requested by Dr. Pekka Lauri in his review of this manuscript.

---

## Author Response (AR1)

We would like to thank Dr. Pekka Lauri and Mr. Walter Cervi Rossi for the time spent on reviewing our manuscript and the valuable remarks which pointed out some important issues which will help to further improve the paper.

Below is our updated point-by-point response to the reviews including a list of all relevant changes made in the manuscript. Reviewer comments are in grey, and our responses are in black. The responses below are an update version of our earlier responses to the reviewers during the open discussion phase of this manuscript from https://doi.org/10.5194/gmd-2021-76-AC1 and https://doi.org/10.5194/gmd-2021-76-AC2

**General comments (Dr. Pekka Lauri):**

p12L200-215: The expansion of timber plantation depends on the share of production that comes from plantations (η). This parameter is exogenous and extrapolated from Pöyry (1999). This means that the expansion of plantation is not endogenous in the model, but it is taken as given. According to figure A5, the model assumes that share plantation increases on average from 25% in 2000 to 62.5 % in 2100. This issue should be made clear already in the abstract because it has significant impact on the results. For example, if the share of plantations were endogenous in the model, then an increasing demand for roundwood would increase the share of plantations relative to natural forests in EUR region. But because the share is exogenous this does not happen, and EUR region is not able adapt higher demand by intensifying their forest management (Figure A6).

Yes, you are right, this is an important point. We will make this limitation clear in the abstract and add a sentence that highlights the assumption regarding the η parameter.

p1L14 in the revised manuscript is changed as follows: (...) elsewhere. *Using exogenous extrapolation of historical roundwood production from plantations and timber demand, we prescribe expansion of forest plantations at the regional level*. As a (...)

The outcome of the optimal rotation models depends much on interest rates and usually these models include sensitivity analysis relative to different interest rate. To avoid this complication, the rotation times could be solved by maximizing increment (f'=f(ac)/ac) instead of maximizing NPV (f'(ac)/f(ac)=r). This would also be more reasonable objective for the recursive dynamic model where all other choices are based on recursive optimization instead of intertemporal optimization.

Thank you for this important comment. We added this capability in the model to calculate rotation lengths via maximization of increment, however we now choose rotation length based on maximization of current annual increment i.e. f'(ac) only.

Justification for this assumption is reflected in p7L133:141 as well as additional discussion on this in p27L400:408 of the revised manuscript.

Add some discussion about the forest age-class dynamics and optimal rotation models in the introduction. Basically move some material from discussion to introduction. Including forest age-class dynamics in the large-scale land-use model is the main contribution of the study, but this issue is completely ignored in the introduction.

We moved the segment discussing the forest age-class dynamics and rotation lengths to the introdzuction in p3L78:81 of the revised manuscript.

**Specific comments (Dr. Pekka Lauri):**

p2L26: According to FAOSTAT global roundwood demand was 3969 Mm3 in 2019 and industrial roundwood 2024 Mm3. Global roundwood demand cannot be 1683 Mm3.

We replaced this instance of *roundwood* with *industrial roundwood* in p2L28 of th revised manuscript.

As there are no new estimates of the share of industrial roundwood production coming from plantations, we assume that the trends observed by Jürgensen et al. 2014 still hold. We also updated the industrial roundwood production data now based on latest FAOSTAT numbers.

p2L28 in the revised manuscript is updated to (...) *likely supply more than 33% (654 Mm3) of global industrial roundwood demand (1984 Mm3) in 2020 based on historical trends (Jürgensen et al., 2014).*

p2L26 Add reference or explanation for 33% (560 Mm3) plantation supply -> Pöyry (1999) extrapolation (Figure A5). This it is not data but model outcome.

We added Jürgensen et al. 2014 as a reference for this sentence.

In fig. A5 the caption has a new formulation as: *Modeled contribution of timber harvest from natural forests and plantations to industrial roundwood and wood fuel production in forestry scenario (1995-2100)..*

p2L38: Add more relevant references for high roundwood productivity of plantations relative to natural forests than FAO (2013), e.g. IPCC (2006). Also, add some explanation why roundwood productivity is higher in plantations than managed natural forests.

We added the following references for high roundwood productivity of plantations relative to natural forests:

a)  IPCC 2006
b)  Payn et al. 2015
c)  Cubbage et al. 2007
d)  Evans and Turnbull 2004

From a qualitative point of view, plantations have more control on breeding material, fertilization, management intensity etc. than managed natural forests and hence more control of quality and quantity.

We added following explanation for higher productivity in plantations than managed natural forests in p2L40 of the revised manuscript: (..)imperative. P*lantation forests for timber production have potentially higher annual average increment per area than natural forests and managed natural forests (IPCC 2006, Payn et al. 2015, Cubbage et al. 2007, Evans et al. 2004) because they are managed more intensively (fertilizer, thinning) and rely on high quality seeds and seedlings for regeneration. Because of* their (...)

**p6L124, p6L132, p7L136: Equation should be f '/ f=r.**

We updated the formulation in eq. 1. with new assumption regarding choice of rotation length.

**Rotation times for timber plantations in Figure 4 are "interesting", but the question is how reasonable they are. For example, with 30-40 years rotation time in Russia and Europe you get only pulpwood (sawlogs require 60-100 years rotation). Moreover, it is not clear why rotation times are longer in North-America than in Europe and Russia. Is this connected to interest rates or productivity? There is only a small difference in interest rates (Table A2) and there should not be large differences in biomass growth between these regions. Some discussion of this should be added and eventually an update to growth curves, interest rate data and add a minimum diameter constraint for sawlogs.**

This is a very important point. We noticed that not having enough heterogeneity in the parametrization of our growth function with underlying parameters from Braakhekke et al. 2019 resulted in relatively homogenous rotation lengths within MAgPIE regions.

We changed our rotation length calculation to maximize current annual increment – decoupling the calculation from dependence on interest rate. Assumptions regarding growth curves in MAgPIE are described in p8L146:147.

In MAgPIE we do not use or model the minimum diameter constraint for sawlogs. Biomass extraction from trees is calculated based on expected yield and area information for simplicity. We added additional discussion regarding this in p27L414:418 of the revised manuscript.

**Is rotation time for natural forests determined by the same rule than for timber plantations (equation 1). If yes, then add similar map (Figure 3) for natural forest rotation time. It would be interesting to see the regional difference between timber plantations and natural forests rotation times. If no, then add some justification why natural forest rotation time is chosen differently than in timber plantations. Basically explain also natural forest rotation lengths in chapter 2.3.**

Natural forests are not bounded by rotation length constraints. The model is free to choose which age-class in natural forests to harvest based on harvesting costs and associated trade-offs i.e., during each optimization step, while harvesting natural forests, a decision is made whether it would be cheaper to harvest from alternative sources i.e., plantations. MAgPIE's objective function is to minimize global production costs. We use a lower harvesting cost (per ha) for plantations than in natural forests. This implicitly provides a signal to the model to harvest forests with higher growing stock first.

We added an additional sentence in p8L150 of the revised manuscript for clarity: (...) *Natural forests are not bounded by rotational constraints of plantations*.

We renamed section 2.3 to *Rotation lengths* instead.

**According to Figure A8 EUR region growing stock decreases close to zero in 2100, which implies that forest management is not sustainable in EUR region. Easy way to avoid this would be to add additional "sustainability" constraint on harvests (harvests ≤ α x increment where α=1 for normal forests, α> 1 for old forests and α < 1 for younger forests). Alternatively, increase the share of plantation in EUR region. Basically take**

account in extrapolation of η that demand increases in the future. Interesting is also opposite development of growing stock in CHA region. This seems to be connected to the higher share of roundwood production coming from plantation in CHA region (Figure A7). Why cannot EUR region do the same as CHA region and avoid the decrease in growing stock ?

We increased the share of production which can possibly come from plantations in EUR. Altering rotation length calculations by maximizing CAI and increasing the share of production than can in principle come from plantations in EUR (0.54 in 1995 to 0.86 in 2100) now results in stable growing stock development over time in EUR.

Updated numbers are shown in table A4 and updated results are shown in fig. A5, fig. A7 and fig. A8 in the revised manuscript.

**General comments (Mr. Walter Cervi Rossi):**

I understood the principle of rotation lenghts derived from the relation between interest rate and IGR. This generates information on rotation lenghts and it is important for calculating timber production costs. I was wondering whether some reference data could be added to validate the rotation lenght map. I`m not a forest specialist, but in principle Latin America and Australia should have the similar spatial pattern, given that that both regions are important eucalyptus producers. Instead, the map shows different spatial patterns. I believe that an analysis on the main species (and they respective rotation period) per region could be used to calibrate the rotation lenght map.

Thank you for this important comment. Using FAO 2006 (Global planted forests thematic study. Results and Analysis), the mean rotation length in Australia is 34 years (min = 28 years, max = 40 years) - mean rotation length in Latin America (using South America from FAO reported numbers) is 24 years (min = 18 years, max = 29 years). Leech 2013 suggest an optimal rotation of 45 years in Australia. Additionally, the intensity of the management, e.g. using genetic engineered plant material, irrigation and fertilization might differ between Latin America and Australia a lot, even for the same species. As these are aggregated numbers, it cannot be determined with certainty if the spatial patterns on a finer spatial scale (as in fig. 4) are going to be similar between Latin America and Australia.

We have added an additional panel in fig. 4 of the revised manuscript, using the numbers from FAO 2006 to validate the rotation lengths from MAgPIE.

MAgPIE in its current format cannot handle tree species information. Calculation of rotation lengths at the cellular level is one of the novelties of this manuscript. Calibration of cellular rotation lengths to a regional or country level data-set would result in loss of the spatial (cellular) level differentiation in rotation lengths. Instead, for the initial time step, we calibrate the growing stocks to FRA 2020 reported numbers for both natural forests and plantations. Also using a single value for rotation length per MAgPIE region would not be ideal as there are spatial differences in the way plantations grow within each region. Our way of deriving rotation lengths based on carbon stock information from LPJmL for natural vegetation (different species dominate in different cells) helps us in using spatial differences in carbon densities to act as a proxy for differences in species. We are also not aware of any spatially explicit data which is on a finer spatial scale (e.g., 0.5° resolution) with information on both

tree species and associated rotation length which can be used to calibrate MAgPIE's cellular level rotation lengths correctly.

We added additional discussion regarding this in p27L400:408 and p27L414:418 of the revised manuscript.

*It is clear that the study was focused on creating a proof of concept that enables a forestry module in MAGPIE. However, I missed a bit the discussion on how realistic are the figures presented. For example, in section 3.1, it is mentioned a large increase in cropland at expense of primary forest areas (where exactly?). Are protected areas included in the analysis? What about agro-forestry (also included?)? Perhaps some discussions around the current and future spatial uncertainties would be relevant as well.*

We added a regional level figure (fig. A9) in the appendix to clarify where does cropland expand at the cost of primary forests (and other land-use types).

In terms of protected areas, the manuscript accounts for National Policies Implemented (NPIs) in terms of forest protection and afforestation according to existing national policies until 2030, in support of the Paris Agreement. Additional land protection is based on The World Database on Protected Areas (WDPA) which earmarks category I and II areas from International Union for Conservation of Nature (IUCN) as protected in MAgPIE.

This is now made clear in the method section with additional paragraph in p7L128:131 as well as in Table 1 with an additional column.

Agroforestry is not included in MAgPIE.

In p27L419:p28L426 of the revised manuscript we added further discussion on how realistic the figures are based on uncertainties in the socio-economic model drivers across SSPs as this manuscript only considers the SSP2 scenario. We also added further discussion about the uncertainty of spatially explicit data on plantation forest, with respect to the differentiation between productive and non-productive plantations – which in turn also has a bearing on the results in this manuscript and that the management of plantations also depends on other factors such as availability of workforce, investment, R&D available to improve the management etc.

*Productivity is an important component for calculating global costs of demand driven land uses. I`m not fully aware of LPJML, but doing a quick research, I found that LPJML might include the productivity of natural vegetation and planted forests. These were surely incoporated in MAGPIE (right?). But yet it is not clear to me how the productivity of secondary forest came about?*

For every cell, the long-term carbon density of natural vegetation from LPJmL is converted to age-class dependent carbon densities by a Chapman-Richards volume growth function. Based on information from LPJmL, we use different parameters in the Chapman-Richards function for plantations and natural vegetation, resulting in faster regrowth of plantations as compared to natural vegetation. The resulting age-class dependent carbon densities for plantations and natural vegetation are converted to harvestable yield with the help of biomass conversion and expansion factors. For secondary forests, we use these age-class specific harvestable yields. We initialize the age-class structure of secondary forest based on observational data (Poulter et al. 2019). For primary forests, we use the highest age-class yield, which reflects natural

vegetation. This is implicitly defined in p10L166:167. We will further clarify it with an additional sentence.

We further clarified it with an additional sentence in p12L211: (...) *y is the age-class (ac) and forest type specific biomass yield in tDM/ha*, C is the (...).

To avoid confusion, we changed p2L34 of the revised manuscript to make it clear that we include secondary forest (and primary forest) in our definition of the term "natural forest", based on the rationale that secondary forest is regrown natural forest.

p13L215 in revised manuscript is updated as: (...) et al. (2020a). *Harvestable biomass yield (yr) is different between natural forests(primary and secondary forests) and plantations by virtue of differences in parametrization of underlying growth function(s).Primary forests are assumed to exist in highest age-class hence are attributed with old-growth forest yields. Both, secondaryforests and plantations yields are age-class specific but differ in growth-dynamics.*

**Specific comments (Mr. Walter Cervi Rossi):**

Line 175. If a fraction of forestry residues is recovered during the harvesting period, it is likely that there will be a potential decay in productivity in the forthcoming period in comparison with a plantation system that does not recover any fraction of residues. It was not cleat for me if that was included in the model, but it is something to be considered

In p11L196:p12L200 of the revised manuscript, we added an additional paragraph in methods section to include these assumptions.

**Typo (Mr. Walter Cervi Rossi):**

Line 119. "optimal"

The line containing this typo will be removed as we will re-write the rotation length calculation segment as requested by Dr. Pekka Lauri's review.